# Distinct mechanisms for TMPRSS2 expression explain organ-specific inhibition of SARS-CoV-2 infection by enzalutamide

Fei Li[1,2,8], Ming Han[1,2,8], Pengfei Dai[1,2,8], Wei Xu[3,8], Juan He[1,2,8], Xiaoting Tao[4,5,8], Yang Wu[3,8], Xinyuan Tong[1,2], Xinyi Xia[1,2], Wangxin Guo[1,2], Yunjiao Zhou[3], Yunguang Li[1], Yiqin Zhu[1], Xiaoyu Zhang[1], Zhuang Liu[1], Rebiguli Aji[1,2], Xia Cai[3], Yutang Li[3], Di Qu[3], Yu Chen [6], Shibo Jiang [3], Qiao Wang[3], Hongbin Ji [1], Youhua Xie [3✉], Yihua Sun [4,5✉], Lu Lu [3✉] & Dong Gao [1,2,7✉]

The coronavirus disease 2019 (COVID-19) pandemic, caused by severe acute respiratory syndrome coronavirus 2 (SARS-CoV-2), has rapidly become a global public health threat. The efficacy of several repurposed drugs has been evaluated in clinical trials. Among these drugs, a second-generation antiandrogen agent, enzalutamide, was proposed because it reduces the expression of transmembrane serine protease 2 (TMPRSS2), a key component mediating SARS-CoV-2-driven entry, in prostate cancer cells. However, definitive evidence for the therapeutic efficacy of enzalutamide in COVID-19 is lacking. Here, we evaluated the antiviral efficacy of enzalutamide in prostate cancer cells, lung cancer cells, human lung organoids and Ad-ACE2-transduced mice. Tmprss2 knockout significantly inhibited SARS-CoV-2 infection in vivo. Enzalutamide effectively inhibited SARS-CoV-2 infection in human prostate cells, however, such antiviral efficacy was lacking in human lung cells and organoids. Accordingly, enzalutamide showed no antiviral activity due to the AR-independent TMPRSS2 expression in mouse and human lung epithelial cells. Moreover, we observed distinct AR binding patterns between prostate cells and lung cells and a lack of direct binding of AR to TMPRSS2 regulatory locus in human lung cells. Thus, our findings do not support the postulated protective role of enzalutamide in treating COVID-19 through reducing TMPRSS2 expression in lung cells.

[1] State Key Laboratory of Cell Biology, Shanghai Key Laboratory of Molecular Andrology, Shanghai Institute of Biochemistry and Cell Biology, CAS Center for Excellence in Molecular Cell Science, Chinese Academy of Sciences, Shanghai 200031, China. [2] University of Chinese Academy of Sciences, Beijing 100049, China. [3] Key Laboratory of Medical Molecular Virology (MOE/NHC/CAMS), School of Basic Medical Sciences and Biosafety Level 3 Laboratory, Shanghai Medical College, Fudan University, Shanghai 200032, China. [4] Department of Thoracic Surgery, Fudan University Shanghai Cancer Center, Shanghai 200032, China. [5] Department of Oncology, Shanghai Medical College, Fudan University, Shanghai 200032, China. [6] Human Oncology and Pathogenesis Program, Memorial Sloan-Kettering Cancer Center, New York, NY 10065, USA. [7] Institute for Stem Cell and Regeneration, Chinese Academy of Sciences, Beijing 100101, China. [8] These authors contributed equally: Fei Li, Ming Han, Pengfei Dai, Wei Xu, Juan He, Xiaoting Tao, Yang Wu. ✉email: yhxie@fudan.edu.cn; Sun_yihua76@hotmail.com; lul@fudan.edu.cn; dong.gao@sibcb.ac.cn

C oronavirus disease 2019 (COVID-19), which is caused by the novel coronavirus severe acute respiratory syndrome coronavirus 2 (SARS-CoV-2), has emerged as a new pandemic. COVID-19 has led to nearly 28,000,000 confirmed global cases and 910,000 deaths as of September 11, 2020[1]. SARS-CoV-2 is a serious worldwide threat due to its high transmissibility[2,3]. The current pandemic and the potential for future pandemics have exposed the urgent need for the rapid development of efficient countermeasures. Therefore, repurposing clinically proven drugs has been postulated as a promising strategy for developing treatments for SARS-CoV-2 infection.

Transmembrane serine protease 2 (TMPRSS2) has been reported with essential role in mediating viruses, including SARS-CoV-2-driven entry into host cells[4–7]. The spike glycoprotein (S) of SARS-CoV-2 and its receptor, angiotensin-converting enzyme 2 (ACE2) have been demonstrated with the function in mediating the attachment of SARS-CoV-2 to host cells[8,9]. Then, the priming of SARS-CoV-2 S protein is processed by TMPRSS2[5]. Moreover, besides SARS-CoV-2, both SARS-CoV, another type of coronaviruses and H1N1, an influenza virus, also employ TMPRSS2 for viral entry[6,10,11]. Since the conserved role of TMPRSS2 in promoting coronaviruses and influenza viruses-driven entry into host cells has been highlighted, modulating TMPRSS2 expression or its protease activity is postulated to be a potential method for antiviral intervention[12–15].

Enzalutamide is a potent inhibitor of the androgen receptor (AR) and has been approved for the treatment of castration-resistant prostate cancer (CRPC) patients[16,17]. Mechanistically, enzalutamide binds to AR, reduces the efficiency of its translocation from the cytoplasm to the nucleus, and impairs the AR-mediated signaling pathway[17]. Given the modulation of TMPRSS2 expression by AR in prostate cells[18], several clinical trials have been initiated to assess the therapeutic efficacy of enzalutamide in COVID-19 patients (ClinicalTrials.gov identifiers; NCT04475601 and NCT04456049). However, it remains elusive whether AR indeed controls TMPRSS2 expression in different organs, especially lung. Thus, an investigation of enzalutamide in the treatment of SARS-CoV-2 infection is urgently needed. Herein, we evaluated the antiviral efficacy of enzalutamide in human lung organoids (LuOs) and human ACE2 recombinant adenovirus (Ad-ACE2)-transduced Tmprss2 knockout (Tmprss2-KO) and wild-type (WT) mice. With these powerful approaches, we comprehensively defined the antiviral effect of enzalutamide. Moreover, we showed the potential mechanism of enzalutamide with its different antiviral activity in the human prostate and lung.

## Results

**TMPRSS2 plays a crucial role in promoting SARS-CoV-2 infection.** To elucidate whether TMPRSS2 is crucial for SARS-CoV-2-driven entry into host cells in vivo, we employed previously established Tmprss2-KO mouse model (Supplementary Fig. 1a). In line with previous findings[10,19], under physiological conditions, Tmprss2 knockout exhibited little effect on multiple organs, including lungs (Fig. 1a). In order to identify Tmprss2 positive cells in multiple organs, we next crossed Tmprss2-KO mice with Rosa26-EYFP mice expressing a CAG-driven YFP Cre-reporter (T2Y), when exposed to tamoxifen, this mouse model can be utilized to trace cells of the Tmprss2-positive lineage via detection of YFP expression (Supplementary Fig. 1b). With this model, we further investigated the existence of Tmprss2-positive cells in multiple organs. Notably, in addition to prostate, other essential organs, including lung, kidney, and liver, which are permissive for SARS-CoV-2 infection in human, were characterized with Tmprss2-postive epithelial cells (Fig. 1b and Supplementary Fig. 1c). The broad distribution of Tmprss2-positive cells might indicate the universal function of Tmprss2 in mediating SARS-CoV-2-driven entry in multiple organs.

To confirm the role of Tmprss2 in SARS-CoV-2 infection, we employed a previously reported Ad-ACE2 transduction method to overcome the natural resistance of mice to SARS-CoV-2 infection[20]. Briefly, we first transduced 10–18-week-old WT mice and Tmprss2-KO mice with $2.5 \times 10^9$ PFU of FLAG-tagged Ad-ACE2 adenovirus. Consistent with previous findings, predominant ACE2 expression was observed in the alveolar epithelium, as indicated by FLAG staining (Supplementary Fig. 1d). Five days post Ad-ACE2 transduction, mice were challenged with $1 \times 10^5$ PFU of SARS-CoV-2. Notably, distinct from WT mice challenged with SARS-CoV-2, Tmprss2-KO mice did not exhibit obvious body weight loss (Fig. 1d). Viral loads in the lungs of Tmprss2-KO mice were significantly less than that of WT-mice (Fig. 1e). We also quantified the percentage of lung cells with SARS-CoV-2 infection. Based on the quantification of more than one million cells in 5 mice per group via S protein staining, the percentage of S protein-positive lung cells was significantly lower in Tmprss2-KO mice than in WT mice (Fig. 1f, g). Importantly, compared to WT mice, Tmprss2-KO mice were characterized with less immune cells infiltration, as indicated by the lower percentage of CD45-positive cells in total lung cells (Fig. 1h–j). Compared with the lungs of WT mice challenged with SARS-CoV-2, we found the reduction in mRNA expression of *Il6*, *Cxcl10*, *Ifnb*, and *Ifng* in Tmprss2-KO mice, corroborating less inflammatory responses (Fig. 1k–n). In addition, through integrating epithelial damage, edema, hemorrhage, parenchymal wall expansion, and inflammatory cells infiltration into H&E scores as previously described, we also assessed lung pathology of SARS-COV-2-infected Tmprss2-KO mice and WT mice. We demonstrated milder lung damage in Tmprss2-KO mice than WT mice (Fig. 1o). Collectively, these findings suggested that the lack of TMPRSS2 had effects on SARS-CoV-2 infection, highlighting the important role of TMPRSS2 in mediating SARS-CoV-2-driven entry into host cells.

**Enzalutamide effectively inhibits SARS-CoV-2-driven entry into prostate cells.** Since TMPRSS2 expression is modulated by AR in prostate cells, which promoted us to identify whether AR inhibition can prevent SARS-CoV-2 infection through reducing TMPRSS2 expression. We first surveyed TMPRSS2 and AR expression across a panel of well-characterized prostate cancer cell lines and two previously established organoid lines MSKPCa1 and MSKPCa3[21]. Notably, both qRT-PCR and western blotting demonstrated high AR and TMPRSS2 expression in LNCaP and VCaP cells (Supplementary Fig. 2a, b). Consistent with previous findings[22,23], a marked reduction in TMPRSS2 protein and mRNA expression was induced by AR inhibition using enzalutamide treatment and was validated in both LNCaP and VCaP cells (Supplementary Fig. 2c–f).

For sensitive and convenient detection of SARS-CoV-2-driven entry into host cells, we employed a pseudovirus system by incorporating SARS-CoV-2 S protein and luciferase into pseudoviral particles through cotransfection of pNL4-3.luc.RE and PCDNA3.1 encoding the SARS-CoV-2 S protein. Thus, this system allowed the sensitive detection of SARS-CoV-2 pseudotype entry by measuring luciferase activity. The constructed pseudovirus was named SARS-CoV-2-S. We first asked whether LNCaP and VCaP were susceptible to SARS-CoV-2-S-driven entry. Since undetectable ACE2 expression in LNCaP and VCaP cells was identified, we observed lack of robust SARS-CoV-2-S-driven entry into these cells, as expected (Supplementary Fig. 3b, d). To enable the permissiveness of LNCaP and VCaP cells, we next transduced Ad-ACE2 into these cells (Supplementary Fig. 3a, c).

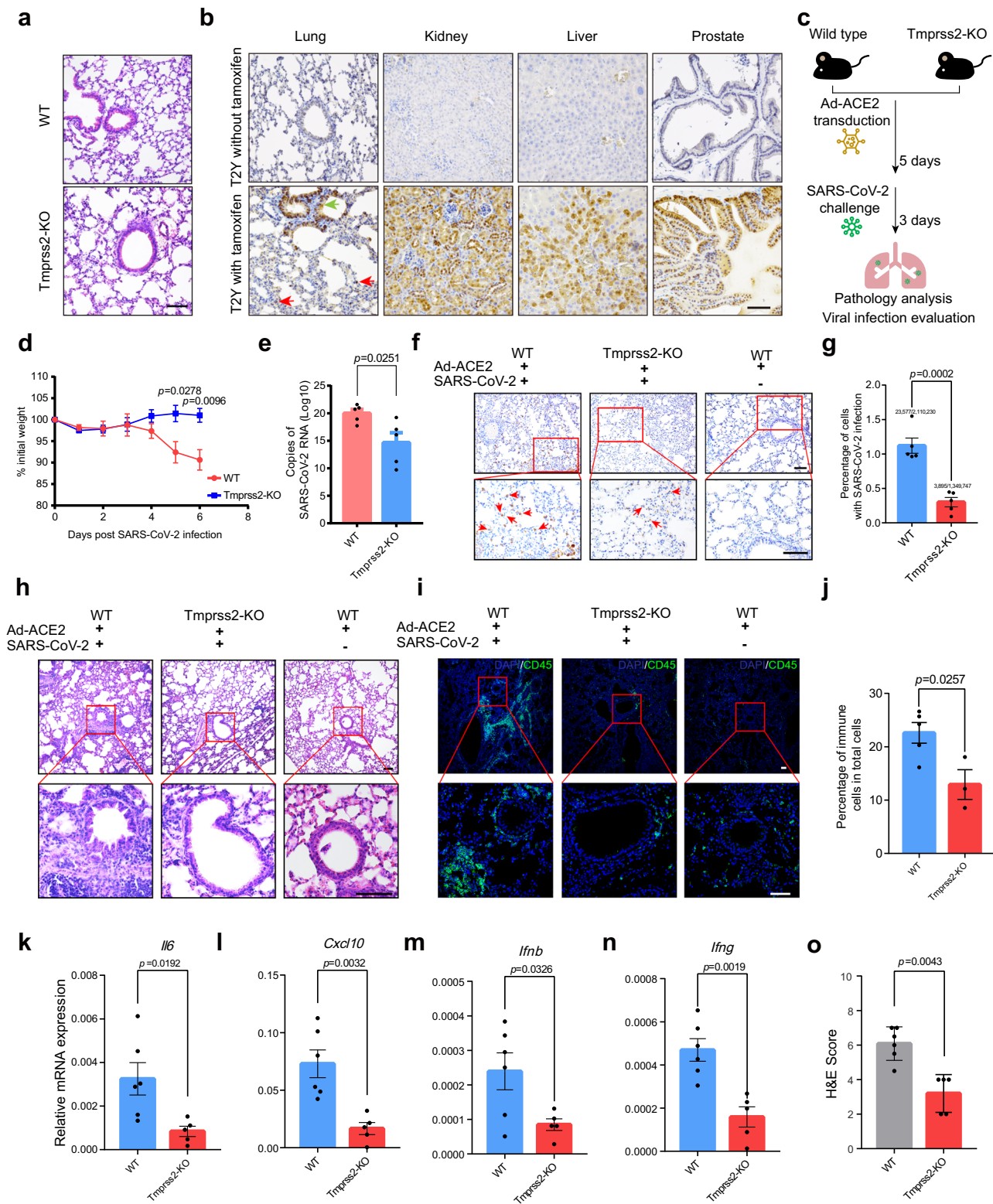

Given that enzalutamide treatment can reduce TMPRSS2 expression in prostate cells[17,23], we next sought to ascertain whether enzalutamide can prevent SARS-CoV-2 from infecting prostate cells through downregulation of TMPRSS2 expression. We first investigated the therapeutic efficacy of enzalutamide in blocking SARS-CoV-2-S-driven entry into LNCaP cells. In line with previous results generated from other TMPRSS2 positive cells[5,24], camostat mesylate, a clinically proven inhibitor for serine protease including TMPRSS2, significantly attenuated the infection of SARS-CoV-2-S, as indicated by the reduction in luciferase activity, suggesting that TMPRSS2 is also an important factor for facilitating SARS-CoV-2-driven entry into LNCaP cells (Fig. 2a). Remarkably, enzalutamide also significantly blocked SARS-CoV-2-S infection, which even exhibited much higher treatment efficacy than camostat mesylate (Fig. 2a). In addition, immuno-fluorescence staining for luciferase also demonstrated the

**Fig. 1 Ad-ACE2-transduced Tmprss2-KO mice demonstrates TMPRSS2 as an important factor for SARS-CoV-2 infection. a** H&E staining of WT and Tmprss2-KO mouse lungs. **b** YFP IHC staining for lungs, kidneys, livers, and prostates of T2Y mice with (bottom) or without (top) tamoxifen administration. Red arrow indicates alveoli cells and green arrow indicates bronchiole cells, respectively. **c** Experimental strategy to identify the role of TMPRSS2 in mediating SARS-CoV-2 infection utilizing Ad-ACE2-transduced mouse models. **d** The percentage of body weight post SARS-CoV-2 infection in initial body weight (two-tailed *t*-test, mean ± SEM, *n* = 7 (day 0), *n* = 7 (day 1), *n* = 4 (day 2), *n* = 3 (day 3), *n* = 3 (day 4), *n* = 3 (day 5), and *n* = 3 (day 6) biologically independent mice for wild type group, *n* = 10 (day 0), *n* = 10 (day 1), *n* = 5 (day 2), *n* = 5 (day 3), *n* = 5 (day 4), *n* = 5 (day 5), and *n* = 5 (day 6) biologically independent mice for Tmprss2-KO group). **e** Viral loads of SARS-CoV-2 in the lungs of WT and Tmprss2-KO mice (two-tailed *t*-test, mean ± SEM, *n* = 5 biologically independent mice). **f** S protein IHC staining for lungs of Ad-ACE2-transduced wild type mice with (left) or without (right) SARS-CoV-2 challenge and Ad-ACE2-transduced Tmprss2-KO mice (middle) with SARS-CoV-2 challenge, respectively. **g** Quantification of lung cells with SARS-CoV-2 infection indicated by S protein IHC staining (two-tailed *t*-test, mean ± SEM, *n* = 5 biologically independent mice). **h** H&E staining of lungs from Ad-ACE2-transduced WT mice challenged with (left) or without (right) SARS-CoV-2 and Ad-ACE2-transduced Tmprss2-KO mice (middle) challenged with SARS-CoV-2, respectively. **i** Immunofluorescence staining for CD45 in the lungs of Ad-ACE2-transduced wild type mice with (left) or without (right) SARS-CoV-2 challenge and Ad-ACE2-transduced Tmprss2-KO mice (middle) with SARS-CoV-2 challenge, respectively. **j** Quantification for CD45-positive cells in total cells (two-tailed *t*-test, mean ± SEM, *n* = 5 biologically independent mice for wild type group and *n* = 3 biologically independent mice for Tmprss2-KO group). **k**–**n** qRT-PCR analyses on mRNA expression of *Il6* (**k**), *Cxcl10* (**l**), *Ifnb* (**m**), *and Ifng* (**n**) in the lungs of WT and Tmprss2-KO mice challenged by SARS-CoV-2 (two-tailed *t*-test, mean ± SEM, *n* = 6 biologically independent mice for wild type group and *n* = 5 biologically independent mice for Tmprss2-KO group respectively). **o** H&E score quantification for lung lesions in WT mice and Tmprss2-KO mice (Mann–Whitney test, mean ± SEM, *n* = 6 biologically independent mice for wild type group and *n* = 5 biologically independent mice for Tmprss2-KO group). Scale bars represent 50 μm.

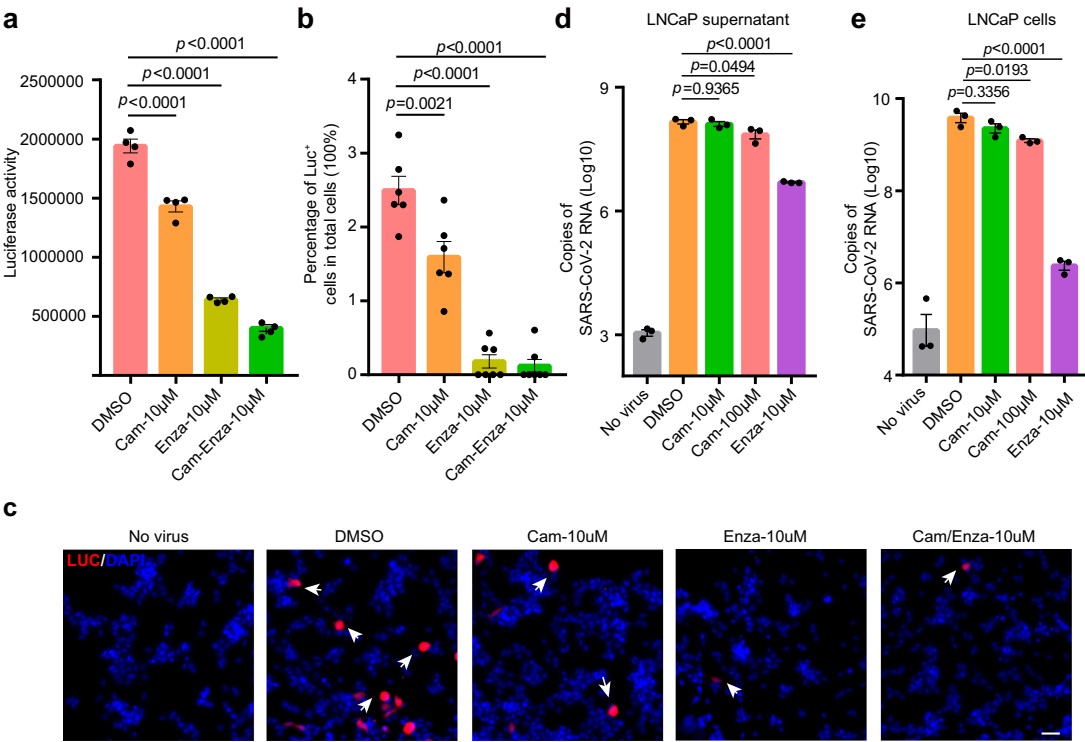

**Fig. 2 Enzalutamide inhibits infection of prostate cells with both authentic SARS-CoV-2 and SARS-CoV-2-S pseudovirus. a** SARS-CoV-2-S-driven entry into LNCaP cells transduced with Ad-ACE2 and treated with DMSO, 10 μM camostat mesylate, 10 μM enzalutamide or a combination of 10 μM camostat mesylate/enzalutamide. Luciferase activity was measured 48 h post SARS-CoV-2-S infection (one-way ANOVA and Tukey's test, mean ± SEM, *n* = 4 biologically independent samples). **b**, **c** Quantification analysis of luciferase-positive cells (**b**) and immunofluorescence of luciferase (**c**) in LNCaP cells transduced with Ad-ACE2 and treated with DMSO, 10 μM camostat mesylate, 10 μM enzalutamide or a combination of 10 μM camostat mesylate/enzalutamide (one-way ANOVA and Tukey's test, mean ± SEM, *n* = 6 biologically independent samples). **d** Copies of SARS-CoV-2 RNA in LNCaP culture medium supernatant with or without SARS-CoV-2 infection and treated with DMSO, 10 μM camostat, 100 μM camostat or 10 μM enzalutamide treatment condition (one-way ANOVA and Tukey's test, mean ± SEM, *n* = 3 biologically independent samples). **e** Copies of SARS-CoV-2 RNA in LNCaP cells with or without SARS-CoV-2 infection and treated with DMSO, 10 μM camostat, 100 μM camostat or 10 μM enzalutamide treatment condition (one-way ANOVA and Tukey's test, mean ± SEM, *n* = 3 biologically independent samples). Scale bars represent 50 μm.

consistent results that enzalutamide significantly reduced the percentage of cells with SARS-CoV-2-S-driven entry (Fig. 2b, c). Since pseudovirus system was limited to investigations on SARS-CoV-2-S-driven entry into host cells, we next assessed whether enzalutamide interferes with authentic SARS-CoV-2-driven entry and the subsequent steps of the viral replication cycle. Consistent with findings from the pseudovirus system, enzalutamide

efficiently exerted antiviral activity against SARS-CoV-2 in LNCaP cells, as demonstrated by the significantly reduced viral titers of SARS-CoV-2 in both culture medium supernatant and cellular contents (Fig. 2d, e). Moreover, we evaluated whether enzalutamide can prevent SARS-CoV-2-S-driven entry into VCaP cells. Recapitulating results in LNCaP cells, enzalutamide treatment also significantly blocked SARS-CoV-2-S-driven entry

into VCaP cells (Supplementary Fig. 3e). Taken together, utilizing both the pseudovirus system and authentic SARS-CoV-2, we demonstrated that enzalutamide efficiently prevented SARS-CoV-2-driven entry into prostate cells by inhibiting AR to reduce TMPRSS2 expression.

**Enzalutamide fails to prevent SARS-CoV-2 from infecting human lung organoids and cells**. Since enzalutamide efficiently inhibited infection of human prostate cells with SARS-CoV-2, we next sought to evaluate its therapeutic efficacy in human lung cells. To this end, we surveyed single-cell RNA-sequencing data from healthy human lungs[25], consistent with previous results[26], TMPRSS2-positive cells were broadly distributed in various cell types, potentially indicating an important role of TMPRSS2 in mediating SARS-CoV-2 infection in multiple lung cell types (Supplementary Fig. 4a, b, d). In particular, high expression levels of both TMPRSS2 and ACE2 were identified in SFTPC-positive alveolar type II (ATII) cells, potentially indicating TMPRSS2-dependent entry of SARS-CoV-2 into these cells (Supplementary Fig. 4a, c, e, f). We further assessed AR expression and found that similar to TMPRSS2, AR was also highly expressed with a wide distribution (Supplementary Fig. 4a, g). These results might indicate the presence of AR-positive cells and TMPRSS2-positive cells in human lungs. To further confirm and quantify the presence of TMPRSS2-positive cells and AR-positive cells, we next performed co-staining immunofluorescence to detect TMPRSS2 and AR expression in normal human lungs. In accordance with the single-cell RNA-seq data, both AR-positive cells and TMPRSS2-positive cells were widely distributed in both bronchiolar and alveolar regions (Supplementary Fig. 4h, i). In alveolar regions, the percentage of AR-positive cells and TMPRSS2-positive cells were 11.10% and 6.00%, respectively, and 1.05% of cells were TMPRSS2/AR double positive cells (Supplementary Fig. 4m, n). Intriguingly, in accordance with results obtained from analyses on single-cell RNA-sequencing data (Supplementary Fig. 4d, g, i, l), higher percentage of both AR-positive cells and TMPRSS2-positive cells was observed in bronchiolar regions. 66.08% and 51.21% of cells were identified as AR-positive cells and TMPRSS2-positive cells, respectively, while, the percentage of TMPRSS2/AR double positive cells was 37.18% (Supplementary Fig. 4m, o). Collectively, these results clearly demonstrated the existence of TMPRSS2/AR double positive cells in human lungs, as well as indicating the differential percentage of TMPRSS2/AR double positive cells in bronchiolar and alveolar regions.

Since human lungs were characterized with both AR and TMPRSS2 expression, we next sought to determine whether AR can modulate TMPRSS2 expression in the lungs. We firstly established human lung organoids (LuOs) derived from adjacent normal lung tissues with similar culture protocol as previously reported[27] (Fig. 3a). To verify whether LuOs are an appropriate model in which to evaluate the therapeutic efficacy of enzalutamide, we performed immunofluorescence staining for AR and TMPRSS2 in LuOs. By staining of serial sections, we identified both AR expression and TMPRSS2 expression in LuOs, which also contained AR/TMPRSS2 double-positive cells (Fig. 3b). We next employed LuOs to explore whether enzalutamide could manipulate TMPRSS2 expression. Distinct from the above findings in prostate LNCaP cells, enzalutamide treatment did not significantly reduce mRNA expression of *TMPRSS2*, validated in three LuOs lines (Fig. 3c–e). To ensure the permissiveness of LuOs for SARS-CoV-2-S-driven entry, we also transduced these organoids with Ad-ACE2 (Fig. 3f). Twenty-four hours post Ad-ACE2 transduction, LuOs were pretreated with 10 μM camostat mesylate or 10 μM enzalutamide for 48 h before virus infection (Fig. 3g). Camostat mesylate but not enzalutamide inhibited

infection of LuOs with SARS-CoV-2-S, confirming that enzalutamide could not protect lung cells against SARS-CoV-2 infection (Fig. 3h–j). Moreover, we also evaluated whether enzalutamide blocked authentic SARS-CoV-2-driven entry and viral replication. Consistent with the results obtained with the SARS-CoV-2 pseudovirus, enzalutamide did not exhibit antiviral activity against authentic SARS-CoV-2 (Fig. 3k).

Given that lack of treatment efficacy of enzalutamide in blocking SARS-CoV-2-driven entry was characterized in human lung organoids, we also employed multiple lung cancer cell lines to validate whether these results could be recapitulated. Among these cell lines, three of eight, namely, H1437, H2126, and A549 cells were AR-positive, confirming the wide distribution of AR expression across multiple lung cell types (Supplementary Fig. 5a). Since only H1437 and H2126 cells exhibited detectable TMPRSS2 expression, we treated these two cell lines with the AR ligand dihydrotestosterone (DHT) and the AR inhibitor enzalutamide to assess changes in TMPRSS2 expression (Supplementary Fig. 5b). Notably, unlike in LNCaP cells, in which DHT stimulated and enzalutamide reduced TMPRSS2 expression, no obvious changes in mRNA expression of *TMPRSS2* were observed in H2126 and H1437 cells treated with these two agents (Supplementary Fig. 5c–f). In addition, we also performed immunofluorescence for TMPRSS2 to validate these results. Distinct from results in LNCaP cells that DHT stimulated TMPRSS2 expression and enzalutamide reduced TMPRSS2 expression, respectively, no obvious changes in TMPRSS2 expression were observed in H2126 and H1437 cells with these two treatments (Supplementary Fig. 5g). To enable these lung cells to be susceptible to SARS-CoV-2-S-driven entry, we also transduced Ad-ACE2 into these cells (Supplementary Fig. 5h). We next compared camostat mesylate and enzalutamide-induced inhibition of SARS-CoV-2-S entry into H2126 cells (AR-positive and TMPRSS2-positive), H23 cells (AR-negative and TMPRSS2-negative) and Calu-3 cells (AR-negative and TMPRSS2-positive). Both H2126 and Calu-3 cells were dependent on TMPRSS2 for SARS-CoV-2-S infection, as indicated by the inhibitory effects of camostat mesylate. Consistent with the finding that enzalutamide could not reduce TMPRSS2 expression in H2126 cells, enzalutamide did not inhibit infection of either H2126 cells or AR-negative Calu-3 cells by SARS-CoV-2-S (Supplementary Fig. 5i, j). As a control, we also used AR-negative and TMPRSS2-negative H23 cells, in which neither camostat mesylate nor enzalutamide exhibited therapeutic efficacy in preventing viral entry (Supplementary Fig. 5k). Intriguingly, we also noticed that H2126 and Calu3 exhibited differential susceptibility to camostat mesylate, indicating their differential dependency on TMPRSS2-initiated SARS-CoV-2 infection. We next examined mRNA expression of other factors in mediating SARS-driven entry into host cells, including *CTSB/L* and *FURIN*[5,28]. Through surveying CCLE (cancer cell line encyclopedia) expression profiles, we observed remarkably higher mRNA expression levels of both *CTSL* and *FURIN* in H2126, potentially indicating TMPRSS2-independent SARS-CoV-2 infection and explaining less efficacy of camostat mesylate in these cells (Supplementary Fig. 5l). Collectively, utilizing both human lung organoids and lung cancer cells, we demonstrated that TMPRSS2 expression was independent of AR expression in human lung epithelial cells, thus AR inhibition using enzalutamide did not reduce TMPRSS2 expression to block SARS-CoV-2-driven entry into human lung epithelial cells.

**Mouse model for COVID-19 demonstrates the lack of therapeutic efficacy of enzalutamide in interfering with SARS-CoV-2-driven entry**. In order to identify whether AR was not capable of modulating TMPRSS2 expression utilizing in vivo

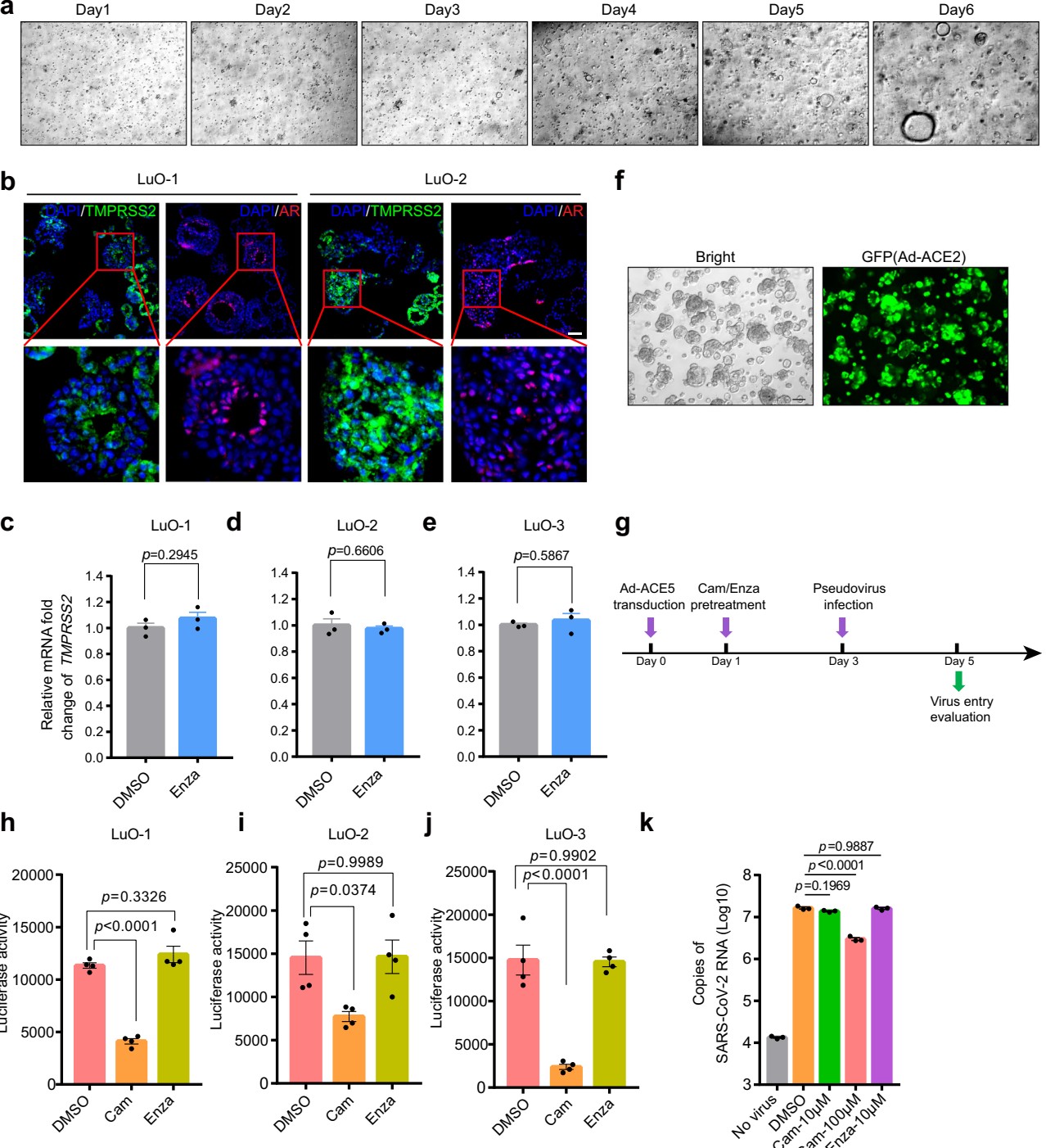

**Fig. 3 Human lung organoids demonstrates lack of efficacy of enzalutamide in inhibiting SARS-CoV-2 infection. a** Lung organoids culture for the first 7 days post dissociation from normal lung tissues. **b** Immunofluorescence staining of TMPRSS2 and AR in adjacent sections of LuO-1 (left) and LuO-2 (right). **c–e** qRT-PCR analysis of *TMPRSS2* mRNA expression in LuO-1 (**c**), LuO-2 (**d**) and LuO-3 (**e**) with enzalutamide treatment for 48 h (two-tailed *t*-test, mean ± SEM, *n* = 3 biologically independent samples). **f** LuOs with Ad-ACE2 expression 48 h post transduction. GFP immunofluorescence represents the adenoviral transduction efficiency. **g** Schematic timeline of evaluating SARS-CoV-2-S-driven entry into LuOs. **h–i** SARS-CoV-2-S-driven entry into LuO-1 (**h**), LuO-2 (**i**) and LuO-3 (**j**) organoids transduced with Ad-ACE2 and treated with DMSO, 10 μM camostat mesylate or 10 μM enzalutamide treatment condition. Luciferase activity was measured 48 h post SARS-CoV-2-S infection (one-way ANOVA and Tukey's test, mean ± SEM, *n* = 4 biologically independent samples). **k** Copies of SARS-CoV-2 RNA in LuOs with or without SARS-CoV-2 infection and treated with DMSO, 10 μM camostat, 100 μM camostat or 10 μM enzalutamide (one-way ANOVA and Tukey's test, mean ± SEM, *n* = 3 biologically independent samples). Scale bars represent 50 μm.

mouse models, we next treated WT mice and castrated mice with enzalutamide for 7 days. Notably, in castrated mice, enzalutamide treatment impaired the function of AR by blocking its nuclear translocation in prostate cells (Supplementary Fig. 6a). As observed in human prostate cells, reduced *Tmprss2* mRNA levels were identified in prostate epithelial cells in enzalutamide-treated mice and castrated WT mice (Fig. 4a). No significant changes in *Tmprss2* mRNA levels in response to enzalutamide treatment and

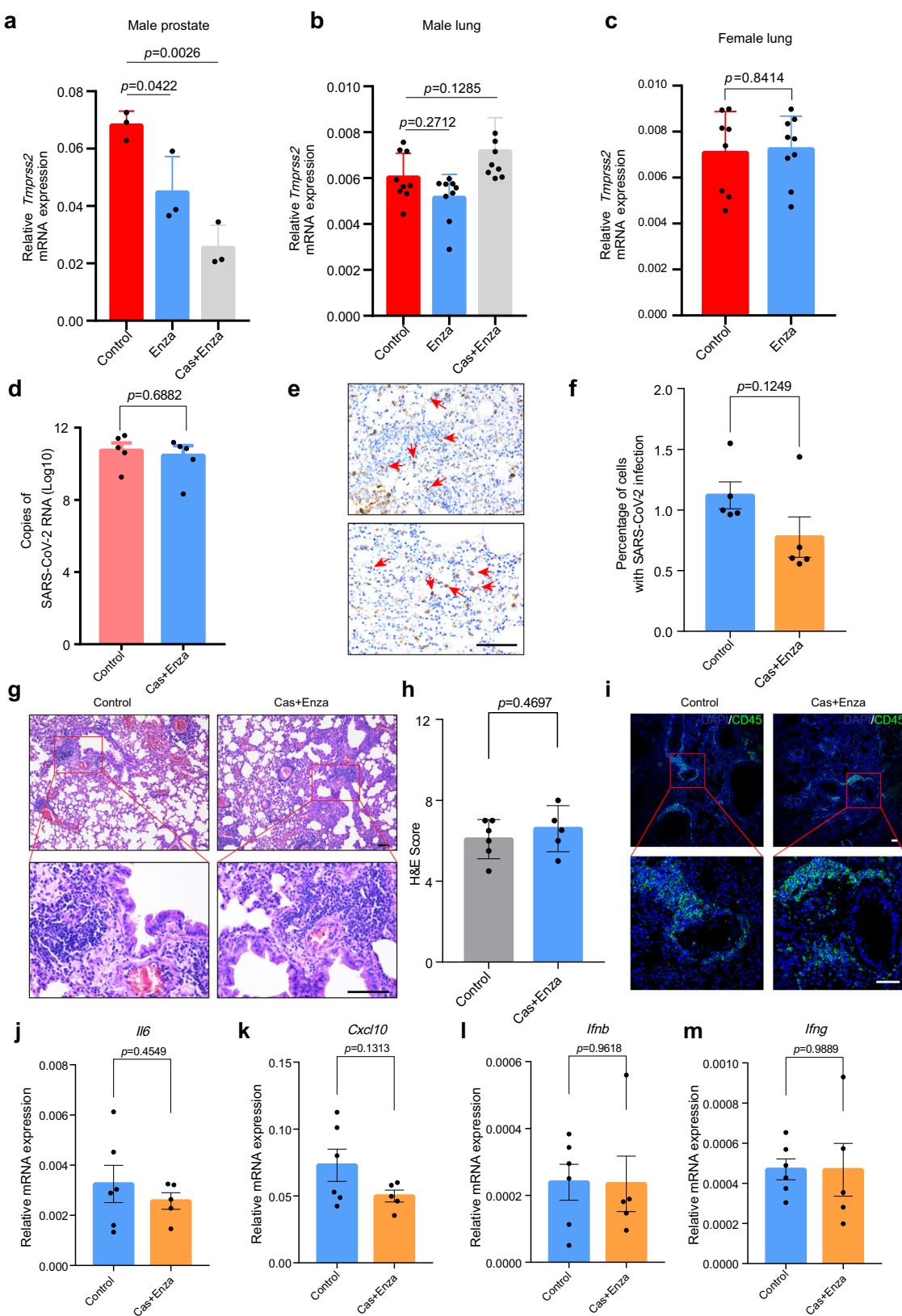

castration were observed in the lungs of male mice (Fig. 4b). In addition, consistent results were obtained in the lungs of female mice treated with enzalutamide (Fig. 4c). Finally, consistent with findings in human prostates and lungs, in vivo experimentation in mice also demonstrated the organ-specific role of AR in regulating TMPRSS2 expression.

To demonstrate enzalutamide treatment efficacy in preventing SARS-CoV-2-driven entry into lung cells utilizing in vivo models, we also employed Ad-ACE2 transduced mouse models (Supplementary Fig. 6b). Briefly, we firstly treated 10–12-week-old wild type C57BL/6 mice with or without enzalutamide treatment by daily intragastric gavage. We next transduced

**Fig. 4 Enzalutamide does not exhibit significant effects on SARS-CoV-2 replication in in vivo Ad-ACE2-transduced mouse models. a, b** qRT-PCR analysis of *Tmprss2* mRNA expression in the prostates (**a**) and lungs (**b**) of male WT control mice, enzalutamide-treated mice and enzalutamide-treated castrated mice (one-way ANOVA and Tukey's test, mean ± SEM, *n* = 3 prostates from 3 biologically independent mice, *n* = 9 lungs from 3 biological biologically independent mice (3 lung lobes per mouse)). **c** qRT-PCR analysis of *Tmprss2* mRNA expression in the lungs of WT control female mice and enzalutamide-treated female mice (two-tailed *t*-test, mean ± SEM, *n* = 8lungs and *n* = 9 lungs for WT control group and enzalutamide-treated group respectively, lung lobes were dissected from 3 biologically independent mice)). **d** Copies of SARS-CoV-2 RNA in the lungs of wild type control mice and enzalutamide-treated castrated mice with SARS-CoV-2 infection (two-tailed *t*-test, mean ± SEM, *n* = 5). Samples were collected 3 days post SARS-CoV-2 challenge. **e** S protein IHC staining for lungs of Ad-ACE2-transduced wild type mice with (bottom) or without (top) castration and enzalutamide pre-treatment before SARS-CoV-2 challenge respectively. **f** Quantification of lung cells with SARS-CoV-2 infection indicated by S protein staining (two-tailed *t*-test, mean ± SEM, *n* = 5 biologically independent mice). **g** Histological analysis of H&E staining in the lungs of control WT mice (left) and enzalutamide-treated castrated mice (right) infected with SARS-CoV-2. **h** H&E score quantification for lung lesions in control mice and enzalutamide-treated castrated mice (two-tailed Mann–Whitney test, mean ± SEM, *n* = 6 biologically independent mice for wild type control group and *n* = 5 biologically independent mice for enzalutamide-treated castrated group). **i** Immunofluorescence staining for CD45 in the lungs of wild type control mice (left) and enzalutamide-treated castrated mice (right) with SARS-CoV-2 infection. **j–m** qRT-PCR analyses on mRNA expression of *Il6* (**j**), *Cxcl10* (**k**), *Ifnb* (**l**), *and Ifng* (**m**) in the lungs of control mice and enzalutamide-treated castrated mice challenged by SARS-CoV-2 (two-tailed *t*-test, mean ± SEM, *n* = 6 or 5 biologically independent mice for wild type control group and enzalutamide-treated castrated group respectively). Scale bars represent 50 μm.

control and enzalutamide-treated mice with $2.5 \times 10^9$ PFU of Ad-ACE2 adenovirus. Five days post Ad-ACE2 transduction, mice were challenged with $1 \times 10^5$ PFU of SARS-CoV-2. Mouse lungs were collected for pathological analysis and viral load determination 3 days post SARS-CoV-2 challenge. Viral loads did not differ significantly between control and enzalutamide-treated mouse lungs (Fig. 4f). In addtion, the percentage of lung cells infected with SARS-CoV-2 in control mice did not differ significantly from that in enzalutamide-treated mice (Fig. 4g, h). Moreover, histopathological analysis revealed similar levels of pathological lesions and inflammatory infiltration in the lungs of control and enzalutamide-treated mice (Fig. 4i–k). We observed similar mRNA expression levels of some canonical inflammatory cytokines and chemokines, corroborating comparable inflammatory responses to SARSCoV-2 infection in control mice and enzalutamide-treated mice (Fig. 4l–o). Taken together, utilizing in vivo mouse models, we obtained consistent results indicating that enzalutamide did not inhibit SARS-CoV-2-driven entry into lung cells and subsequent viral replication.

**Identification of a distinct AR binding pattern between prostate cells and lung cells.** Given the discrepancy between prostate cells and lung cells in the changes in TMPRSS2 expression in response to enzalutamide treatment, we next sought to elucidate whether such discrepancy was attributed to distinct AR binding pattern. We first performed chromatin immunoprecipitation with sequencing (ChIP-seq) on AR in prostate cells LNCaP and assay for transposase-accessible chromatin using sequencing (ATAC-seq) in both prostate cells LNCaP and lung cells A549, H1437, and H2126. Based on AR ChIP-seq in LNCaP cells, we compared chromatin accessibility among these four cell lines of AR binding sites. Notably, distinct from extensive chromatin accessibility of these sites in LNCaP cells as expected, the other three lung cell lines were characterized with much less open chromatin (Fig. 5a). In principle, transcription factors modulate transcriptional regulation through binding to regulatory elements of target genes, which tightly associates with chromatin accessibility[29]. The chromatin accessibility disparity might indicate distinct AR binding pattern between prostate cells and lung cells. To further characterize AR binding pattern in prostate cells and lung cells respectively, we categorized these AR binding sites into two main groups: "both-open" sites were characterized with open chromatin in both prostate cells LNCaP and the other three lung cell lines (Fig. 5b, Supplementary Data 1), and "prostate-open" sites were identified by the specific existence of open chromatin in prostate LNCaP cells (Fig. 5c, Supplementary Data 2). In order to validate whether chromatin accessibility in AR binding sites

remarkably coordinated AR binding activity, we also performed AR ChIP-seq in the other three lung cell lines. In accordance with open chromatin of a both-open site in *PARP14*, we also observed AR binding in this region in both lung cells and prostate cells (Fig. 5d). Different from this both-open site, upon close inspections on *TMPRSS2*, we only observed AR binding sites in upstream region of *TMPRSS2* in LNCaP cells instead of the other three lung cell lines (Fig. 5e). Compatible with AR binding in *TMPRSS2*, two extra AR binding sites were verified with extensive chromatin accessibility in LNCaP cell but not in other lung cells (Fig. 5e). In addition, unlike in prostate cells, ChIP-qPCR demonstrated the lack of robust AR binding in the upstream region of *TMPRSS2* locus in lung cells (Fig. 5f). These results indicated lack of AR binding in TMPRSS2 in lung cells, which coincided with the above findings that AR inhibition utilizing enzalutamide did not reduce TMPRSS2 expression to inhibit SARS-CoV-2-driven entry. Furthermore, gene set enrichment analysis (GSEA) revealed that androgen response genes were significantly enriched in AR-positive prostate cells (LNCaP, VCaP, and 22RV1) when compared with AR-positive lung cells (A549, H1437, and H2126) (Fig. 5g). In accordance with GSEA results, a significantly higher sum of z-scores for androgen responsive genes was observed in AR-positive prostate cells than that in AR-positive lung cells (Fig. 5h).

Since we demonstrated lack of AR binding in TMPRSS2 in lung cells, utilizing freshly dissociated lung cells from 43 normal human lung tissue samples, we next sought to validate whether the correlation between the expression of *AR* and *TMPRSS2* coordinated these findings. Concordant with above findings, no significant correlation relationship was identified between *AR* and *TMPRSS2* expression (Fig. 5i–k). We also analyzed normal lung tissues and normal prostate tissues from TCGA datasets. A significant and positive correlation between the mRNA expression of *AR* and a both-open gene *PARP14* was observed in both lung and prostate tissues, in keeping with AR binding in this gene (Supplementary Fig. 7a, d). The mRNA levels of both *TMPRSS2* and *FKBP5*, which were characterized with specific AR binding in prostate cells, significantly correlated with *AR* mRNA levels in prostate tissues but not in lung tissues (Supplementary Fig. 7b, c, e, f). These findings established a distinct AR binding pattern between the prostate and the lungs, providing clinical evidence that TMPRSS2 expression is not responsive to AR inhibition in lungs. Collectively, these results revealed a distinct AR binding pattern between human prostate and lung cells. This finding not only offers a mechanistic explanation for the inability of AR to modulate TMPRSS2 expression but also suggests that enzalutamide is not a promising drug for blocking SARS-CoV-2-driven entry into host cells (Fig. 6).

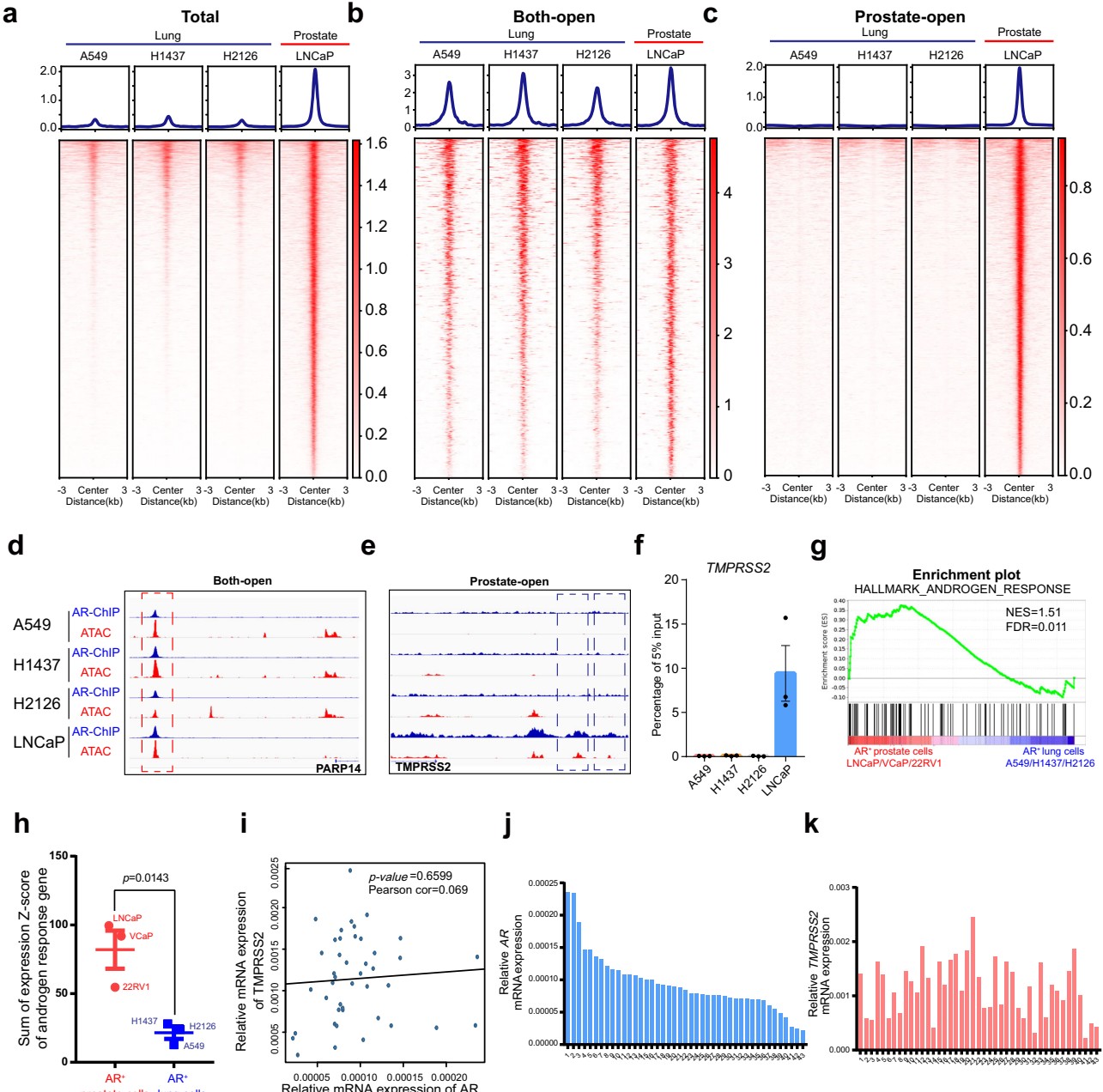

**Fig. 5 Distinct AR binding pattern is characterized between human prostate cells and lung cells. a–c** Profile plot (top) and heatmap (bottom) of ATAC-seq data in a window around the peak center of LNCaP-generated total AR-binding sites (**a**), both-open sites (**b**) and prostate-open sites (**c**) in A549, H1437, H2126, and LNCaP cells, respectively. **d** IGV snapshot representing a both-open peak in *PARP14* locus. **e** IGV snapshot representing a prostate-open peak in *TMPRSS2* locus. **f** ChIP-qPCR analysis of AR binding activity in *TMPRSS2* locus in A549, H1437, H2126, and LNCaP cells, respectively (mean ± SEM, n = 3 biologically independent samples). **g** GSEA enrichment plot of AR-positive prostate cells versus AR-positive lung cells using hallmark androgen response genes. **h** Sum of mRNA expression Z-scores of hallmark androgen response genes in AR-positive prostate cells and AR-positive lung cells (two-tailed *t*-test, mean ± SEM, n = 3 biologically independent cell lines). **i** Correlation analysis for relative mRNA expression of *AR* and *TMPRSS2* in freshly dissociated human lung cells from 43 normal lung tissue samples using qRT-PCR (Pearson correlation test). **j**, **k** Relative mRNA expression of *AR* (**j**) and *TMPRSS2* (**k**) in freshly dissociated lung cells from 43 normal lung tissues.

## Discussion

TMPRSS2 has been demonstrated with a pivotal role in promoting SARS-CoV-2-driven entry into host cells through facilitating S protein priming via its serine protease activity[4–7,30]. These previous findings suggest that the modulation of TMPRSS2 expression may provide an alternative strategy to treat SARS-CoV-2 infection by blocking viral entry into host cells. It is well known that TMPRSS2 expression is regulated by AR in prostate epithelial cells. Enzalutamide, an AR inhibitor approved for use in

CRPC patients, can reduce TMPRSS2 expression in prostate cancer cells. Thus, enzalutamide has been proposed as a promising repurposed drug to inhibit SARS-CoV-2 infection and subsequent replication, which even provoked the initiation of two clinical trials. Here, we further confirmed the indispensable role of TMPRSS2 in SARS-CoV-2 infection using human ACE2-transduced Tmprss2-KO mice (Fig. 1). Consistently, enzalutamide significantly decreased TMPRSS2 expression and inhibited SARS-CoV-2 infection in human prostate cancer cells (Fig. 2).

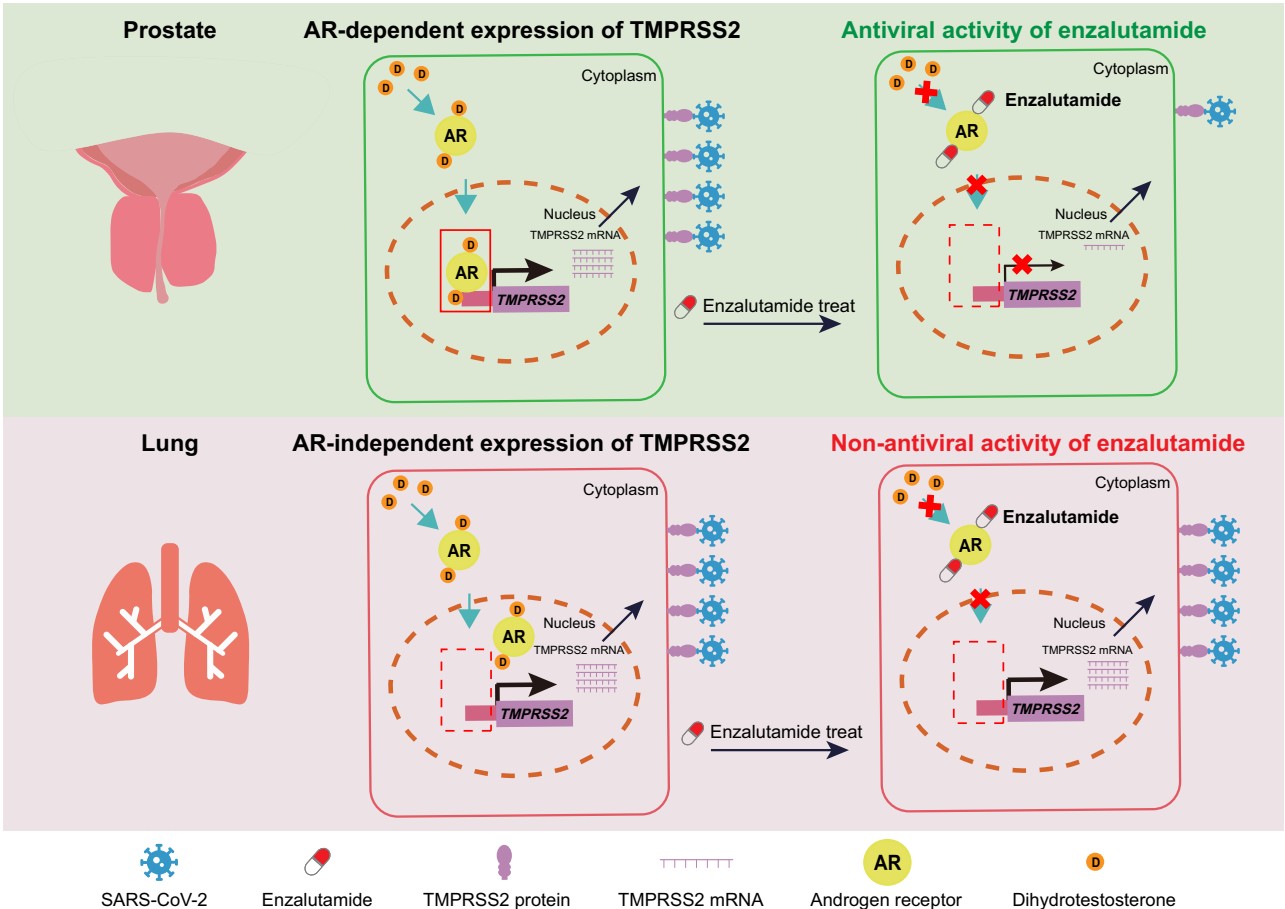

**Fig. 6 Schematic model for distinct mechanisms explaining organ-specific inhibition of SARS-CoV-2 infection by enzalutamide.** In prostate cells, AR can modulate the expression of TMPRSS2, which serves as an important factor in mediating SARS-CoV-2-driven entry. Thus, an AR inhibitor, enzalutamide can reduce TMPRSS2 expression, subsequently resulting in the attenuated entry driven by SARS-CoV-2. However, distinct from its therapeutic efficacy in prostate cells, enzalutamide fails to prevent SARS-CoV-2 infection due to AR-independent TMPRSS2 expression in lung cells.

However, we did not observe any antiviral activity of enzalutamide against SARS-CoV-2 in the lungs of Ad-ACE2-transduced WT mice or human lung organoids. These results suggested that enzalutamide may have antiviral activity in the prostate in male COVID-19 patients, but also indicated that enzalutamide may have no clinical efficacy in treating COVID-19 patients with lung infection.

Enzalutamide reduced TMPRSS2 expression by inhibiting AR activity in LNCaP and VCaP cells, and significantly inhibited infection of these prostate cells by both authentic SARS-CoV-2 or SARS-CoV-2-S pseudovirus. These results indicated that enzalutamide would display effective antiviral activity against SARS-CoV-2 if it could modulate a reduction in TMPRSS2 expression in host cells. Moreover, these findings also highlighted the important role of TMPRSS2 in mediating SARS-CoV-2-driven entry. Given the high efficacy of enzalutamide in inhibiting SARS-CoV-2 in prostate cells, we further assessed its efficacy in human lung cancer cells. Surprisingly, in AR/TMPRSS2 double-positive H2126 and H1437 lung cancer cells, neither AR inhibition using enzalutamide nor AR stimulation using DHT resulted in a significant change in TMPRSS2 expression, implying that AR cannot regulate TMPRSS2 expression in human lung cancer cells. These findings seemed inconsistent with those of previous studies indicating that androgen exposure enhanced TMPRSS2 expression in another lung cell line, A549[31]. This discrepancy might be due to 100 nM testosterone, since such higher concentration of testosterone to treat cells might result in misleading findings,

which could not reflect the physiological function of AR. In addition, notably, TMPRSS2 mRNA was hard to detect under physiological conditions in A549 cells, which exhibit high AR expression (Supplementary Fig. 5b). Given that enzalutamide did not downregulate TMPRSS2 expression in these cells, we further demonstrated that enzalutamide failed to inhibit the entry driven by SARS-CoV-2-S, as expected. Since lung cancer cell lines harbor many genetic alterations, which might lead to disparities in findings with respect to normal lung cells, we employed early-passage benign human LuOs for further study. Compatible with findings in lung cancer cells, enzalutamide had no treatment efficacy in preventing authentic SARS-CoV-2 and SARS-CoV-2-S pseudovirus in benign human lung organoids. Moreover, we also employed Ad-ACE2-transduced mouse models and demonstrated that enzalutamide lacked antiviral activity against SARS-CoV-2 in vivo.

A previous study demonstrated that androgen-deprivation therapy (ADT) significantly reduced the risk of SARS-CoV-2 infection in prostate cancer patients[32]. However, a subsequent study demonstrated that the lethality rate of SARS-CoV-2 in metastatic prostate cancer patients with ADT was not lower than that in other cohorts of infected Italian male patients[33], which did not suggest that ADT exhibited antiviral activity against SARS-CoV-2 in patients with metastatic prostate cancer. The inconsistent findings from these two studies might be attributed to the different populations selected for investigation. However, to date, no concordant and definitive clinical evidence indicates that

ADT, including enzalutamide treatment, significantly inhibits SARS-CoV-2 infection.

Utilizing multiple models, including human lung cancer cells, human lung organoids and Ad-ACE2-transduced WT mice, we demonstrated that enzalutamide failed to inhibit SARS-CoV-2 infection, which was attributed to the lack of AR-driven modulation of TMPRSS2 expression in lung epithelial cells. To elucidate the mechanisms underlying the disparity of AR-driven regulation between prostates and lungs, we further performed AR ChIP-seq and ATAC-seq in AR-positive lung cells and AR-positive prostate cells. Unlike in prostate cells, the lack of specific AR binding at TMPRSS2 locus in lung cells, as demonstrated by AR ChIP-seq, were consistent with the finding that TMPRSS2 expression was independent of AR expression in human lung epithelial cells. These findings indicated mechanisms explaining that the lack of antiviral activity of enzalutamide against SARS-CoV-2 is due to a lack of direct AR binding at the TMPRSS2 locus in lung epithelial cells.

However, our study had limitations. Microenvironmental components, including immune cells, nerve cells and stromal cells, are involved in viral infection and subsequent replication[34–36]. Although we employed Ad-ACE2-transduced in vivo mouse models, our models did not consider the human lung microenvironment. Since stromal cells in multiple human organs, including the lungs, are also characterized by AR expression, we cannot exclude the possibility that enzalutamide might display antiviral activity by altering the expression of some essential cytokines or chemokines in stromal cells.

It was also noting that SARS-CoV-2 could still infect the lungs of Tmprss2-KO mice with lower effectivity. Besides, when transduced with Ad-ACE2, TMPRSS2-negative lung cells H23 was permissive for robust SARS-CoV-2-driven entry (Supplementary Fig. 5k). These findings implied that besides TMPRSS2, other factors may also play a crucial role in promoting SARS-CoV-2 infection. Further studies to identify these factors and their precise functions in mediating SARS-CoV-2 infection will be really necessary.

Estrogen receptor (ER) has been proposed as a potential transcription factor regulating TMPRSS2 expression[13,37]. If so, modulation of ER activity would represent a promising therapy to treat COVID19. However, it still remains unknown that whether ER is indeed able to regulate TMPRSS2 expression in lung cells. To address this question, we first identified two of ER-positive lung cell lines H1437 and H2126 (Supplementary Fig 7a–c), and treated these cells and another ER-positive breast cell lines MCF7 with both ER ligand and inhibitor. Our findings demonstrated that ER could negatively regulate TMPRSS2 expression in breast cells, however, such regulation was lacking in lung cells (Supplementary Fig 7d–g). We also interrogated public ER ChIP-seq and RNA-seq datasets and identified direct binding of ER to TMPRSS2 in breast cells (Supplementary Fig 7h–m). Importantly, based on the mRNA expression of ER and TMPRSS2 in 43 human normal lung samples of our present study and publicly available TCGA datasets, we performed correlation analyses and found no significant correlation (Supplementary Fig 7m–p). These findings might indicate that modulation of ER activity would not be a curative therapy for treating COVID19.

Finally, we took advantage of multiple models of human prostate and lung cells, patients-derived benign lung organoids and Ad-ACE2-transduced Tmprss2-KO and WT mice to comprehensively confirm the pivotal function of TMPRSS2 in SARS-CoV-2 infection. Our findings validated that enzalutamide significantly inhibits SARS-CoV-2 infection in AR and TMPRSS2 double positive prostate cancer cells, identified that enzalutamide does not exhibit antiviral activity in human lung cancer cells and patients-derived benign lung organoids in vitro and in the lungs

of Ad-ACE2-transduced WT mice in vivo, and demonstrated the distinct AR binding pattern between prostate and lung epithelial cells. These findings will enhance our understanding of TMPRSS2 in SARS-CoV-2 infection and indicate the potential failure of clinical trials using enzalutamide to treat COVID-19 patients.

## Methods

**Transduction and infection of mice**. Mice were anesthetized with Avertin (Sigma-Aldrich, T48402-5G) and transduced intranasally with $2.5 \times 10^9$ FFU of Ad5-ACE2 adenovirus in 75 μL DMEM (Gibco C11995500BT). Mice were infected intranasally with $1 \times 10^5$ PFU of SARS-CoV-2 at the fifth day after Ad-ACE2 transduction. Three days post infection, lungs were harvested for virus titer measurement and pathogenicity analysis using qPCR and immunohistology, respectively.

**Study approval**. Mice were generated through standard mouse breeding procedures within the Center for Excellence in Molecular Cell Science (CEMCS) animal facility. All animal experiments were approved by the Institutional Animal Care and Use Committee (IACUC) of CEMCS, and ethical approval was received from the IACUC of CEMCS. Human research was approved by the institutional review board of CEMCS. Informed consent from all participants have been obtained in the present study.

**Castration and enzalutamide treatment of mice**. Enzalutamide (Selleck, S1250) (10 mg/kg; the vehicle contained 1% carboxymethyl cellulose, 0.1% Tween 80, and 5% DMSO) was administered intragastrically to castrated mice daily for 10-30 days according to this study[22].

**Isolation of human lung cells and lung organoid culture**. Non-tumor lung tissue obtained from patients undergoing lung resection was dissected and minced with scissors, washed with 5 mL of DMEM (Gibco, C11995500BT) with Primocin (InvivoGen, ant-pm-2), and then digested with Collagenase II (Gibco, 17101015) for 1–2 h in a cell incubator at 37 °C with shaking. DMEM supplemented with 10% FBS (ExCell (Serum), FSP500) was added to terminate digestion. The suspension was strained through a 70 μm filter, and the cells were then collected by centrifugation at $500 \times g$. If a visible red pellet was produced, erythrocytes were lysed in 8 mL of red blood cell lysis buffer (1 g/L KHCO$_3$, 8.3 g/L NH$_4$Cl, and 0.041 g/L EDTA-Na$_2$.2H$_2$O) for 4 min at room temperature before the addition of 24 mL of PBS and centrifugation at $500 \times g$. Dissociated cells were washed and seeded in growth factor-reduced Matrigel (Corning) and cultured in Advance DMEM/F12 medium supplemented with 10 mM HEPES (Gibco), 2 mM GlutaMAX-1 (Gibco), 500× Primocin (InvivoGen), 1×B27 (Gibco), 1.56 mM N-acetylcysteine (Sigma), 10 mM nicotinamide (Sigma), 0.5 μM A83-01 (Tocris), 10 μM Y27632, 50 ng/mL EGF (Peprotech), 10 ng/mL FGF10 (Peprotech), 1 ng/mL FGF2 (Peprotech), 10% in-house-prepared R-Spondin1, 10% Noggin and 30% Wnt3a.

**qRT-PCR (SYBR)**. Total RNA was extracted with TRIzol reagent (Ambion, 15596018). The solution was mixed well by pipetting several times and lysed at room temperature (RT) for 30–60 min. Then, a 1/5 volume of chloroform was added, and the mixture was vortexed for 15 s. The mixture was then incubated for 2 min and centrifuged at $13,000 \times g$ for 15 min at 4 °C. The aqueous phase was transferred into a new tube, and an equal volume of isopropanol was added. The mixture was centrifuged at $13,000 \times g$ for 10 min at 4 °C. The supernatant was discarded, and the pellet was resuspended in 75% ethanol and centrifuged at $13,000 \times g$ for 7 min at 4 °C. The supernatant was then thoroughly removed and discarded. The pellet was resuspended in 50 μL of nuclease-free water. Reverse transcription was performed with PrimeScriptTM RT Master Mix (TaKaRa, RR036A) with 400 ng of total RNA as input. qRT-PCR was conducted with SYBR qPCR Mix (Qiagen, 208054) using the manufacturer's protocol. The primer sequences are listed in Supplementary Data 3.

**Western blotting**. Cell lysates were prepared in RIPA buffer supplemented with proteinase/phosphatase inhibitors. The protein content was quantified with a BCA (Thermo) assay. Fifteen micrograms of protein were separated via SDS-PAGE and transferred onto a 0.45 mm PVDF membrane (GE). The membrane was blocked for 1 h at room temperature in TBST buffer containing 5% milk and was incubated overnight at 4 °C or for 2 h at room temperature with primary antibodies diluted in TBST buffer containing 5% milk. The membrane was then incubated with rabbit HRP-conjugated secondary antibodies (SAB, #L3012) for 1 h in 5% milk at RT. The primary antibodies included anti-β-Actin (Sigma-Aldrich, A3854, 1:5000), anti-AR (Abcam, ab108341, 1:2000), anti-TMPRSS2 (Abcam, ab92323, 1:1000), and anti-ACE2 (Proteintech, 21115-1-AP, 1:1000), anti-ER-alpha (SANTA CRUZ, sc-543, 1:1000) antibodies.

**Immunohistochemistry and immunofluorescence**. Organoids were fixed with 4% paraformaldehyde for 15 min at room temperature. Fixed organoids were dehydrated sequentially with 95% and 100% ethanol for 10 min, cleared in xylene for 15 min, and then immersed in paraffin 3 times for 30 min each. Human lung

tissues were fixed with 4% paraformaldehyde overnight at 4 °C. The tissue was dehydrated sequentially with 75%, 95%, and 100% ethanol for 1.5 h each, cleared in xylene for 22 min at 55 °C and immersed in paraffin. For immunohistochemistry, freshly sliced 4-micron paraffin sections were subjected to antigen retrieval by boiling for 45 min in 0.01 M citrate buffer. Endogenous peroxidase activity was quenched by immersing the slides in 3% $H_2O_2$ for 20 min. The slides were then blocked in 2% BSA for 1 h and stained with primary antibodies in blocking buffer at 4 °C overnight or at room temperature for 2 h. The slides were then incubated with an HRP-conjugated secondary antibody (OriGene) for 20 min at RT and stained with DAB (Vector Laboratories). The following primary antibodies were used: anti-FLAG (Abmart, M20008F, 1:250) and anti-AR (Abcam, ab108341, 1:200). For immunofluorescence, sections were blocked in 5% goat serum for 1 h at room temperature, stained with primary antibodies at 4 °C overnight, washed with PBST three times and incubated with secondary antibodies for 1 h at room temperature. Sections were then washed with PBST three times and stained with DAPI (Thermo) for 5 min. The monoclonal antibody against the RBD domain of SARS-CoV-2 S protein (anti-S) was cloned from a convalescent SARS-CoV-2 individual and produced by transiently transfecting HEK293F cells. To visualize AR and TMPRSS2 signals, we used Tyramide SuperBoost Kits (Invitrogen, B40926) and incubated the sections for 8.5 min at room temperature. The following primary antibodies were used: anti-SPC (Sigma-Aldrich, ab3786, 1:200), anti-AR (Abcam, ab108341, 1:200), and anti-TMPRSS2 (Abcam, ab92323, 1:250).

**Histopathological analyses.** The criteria to assess the severity of lung damage was modified according to these studies[38,39]. For the degree of bronchiole epithelial cell damage, we scored 0 when no bronchiole epithelial cells damaged, scored 1 when the percentage of damaged bronchiole ducts in total bronchiole reached to less than 10%, scored 2 when the percentage of damaged bronchiole ducts in total bronchiole reached to 10–50%, scored 3 when such percentage was more than 50%. For the degree of degeneration of alveolar epithelial cells, we scored 0 when no alveolar epithelial cells degenerated, scored 1 when the percentage of alveolar epithelial cells in total alveolar epithelial cells reached to less than 10%, scored 2 when the percentage of alveolar epithelial cells in total alveolar epithelial cells reached to 10–50%, scored 3 when such percentage was more than 50%. For the degree of edema, we scored 0 when no edema region was observed, scored 1 when the percentage of edema regions reached to less than 10%, scored 2 when the percentage of edema regions reached to 10–50%, scored 3 when such percentage was more than 50%. For the degree of hemorrhage, we scored 0 when no hemorrhage region was observed, scored 1 when the percentage of hemorrhage regions reached to less than 10%, scored 2 when the percentage of hemorrhage regions reached to 10–50%, scored 3 when such percentage was more than 50%. For the degree parenchymal wall expansion, we scored 0 when no parenchymal wall expansion was observed, scored 1 when the percentage of parenchymal wall expansion regions reached to less than 10%, scored 2 when the percentage of parenchymal wall expansion regions reached to 10–50%, scored 3 when such percentage was more than 50%. For the degree of inflammatory cells infiltration, we scored 0 when no focal inflammatory cells infiltration was observed, scored 1 when the percentage of focal inflammatory cells infiltration reached to less than 10%, scored 2 when the percentage of focal inflammatory cells infiltration reached to 10–50%, scored 3 when such percentage was more than 50%. A H&E score for per mice the sum of scores in every evaluation index. The average of H&E scores from five to six mice for each group was taken as the final H&E score for that group.

**Transduction of cells with Ad-ACE2.** Cells were seeded in 6-well plates before transduction. The next day, Ad-ACE2 was transduced into cells at a multiplicity of infection (MOI) of 100 with polybrene. The culture medium supernatant was replaced with fresh medium 12 h post transduction.

**Pseudovirus production.** SARS-CoV-2 pseudovirus was produced by cotransfection of 293T cells with pNL4-3.luc.RE and PCDNA3.1 encoding the SARS-CoV-2 S protein using Vigofect transfection reagent (Vigorous Biotechnology, T001). One hour before transfection, the medium was replaced with fresh DMEM (GIBCO, C11995500BT). Further transfection was performed according to the manufacturer's protocol. The supernatants were harvested at 48 h post transfection, filtered through a 0.45 μm cell strainer, and split into 1.5 mL tubes for storage at −80 °C.

**Pseudovirus infection assay.** Cells transduced with or without Ad-ACE2 were seeded in 96-well plates at initial count of between 15,000 and 20,000 cells per well and treated with different agents (10 μM camostat mesylate (Selleck, S2874) or 10 μM enzalutamide (Selleck, S1250)). Two days post seeding and treatment, cells were incubated with pseudovirus for 12 h. The culture medium supernatant was then replaced with fresh medium. Two days post virus infection, the culture medium supernatant was removed, and the cells were washed with PBS. The cells were then lysed with 40 μL of 1× Cell Culture Lysis Reagent (Promega, E153A) for 40 min with shaking at 350 × g. Lysis buffer (20 μL) was used for luciferase activity measurement with the Luciferase Assay System (Promega, E151A) in a BioTek Synergy 2 reader.

**SARS-CoV-2 infection.** The SARS-CoV-2 isolate was obtained from a clinical case in Shanghai, China (SARS-CoV-2/SH01/human/2020/CHN, GenBank ID; MT121215). SARS-CoV-2 were propagated in Vero cells. Ad-ACE2-transduced mice were infected intranasally with $1 \times 10^5$ PFU of SARS-CoV-2. Cells and LuOs were infected with SARS-CoV-2 at M.O.I = 0.01 and M.O.I = 1, respectively. All experiments using authentic SARS-CoV-2 were performed in a biosafety level 3 (BSL3) facility at Fudan University.

**qRT-PCR (Probe).** Viral RNA was extracted using TRIzol®LS Reagent (Invitrogen, 10296010) as the manufacture's instruction. One-Step PrimeScript RT-PCR Kit (Takara, RR064) was utilized for qRT-PCR (probe) to measure viral loads with program setting as followed: 95 °C 10 s, 42 °C 5 min for reverse transcription; (95 °C 5 s, 56 °C 30 s, 72 °C 30 s) × 40 cycles for PCR reaction. Sequences of primers are listed in Supplementary Data 3.

**ChIP-seq library preparation.** ChIP-seq was performed as following steps[40]. 10 million cells were fixed with 1% formaldehyde at room temperature for 10 min with rotation. Then, 125 mM glycine was added to quench the formaldehyde at room temperature for 5 min. Cells were washed with cold PBS. The supernatants were then removed, and 880 μL of ice-cold cell lysis buffer (1% SDS, 10 mM EDTA, 50 mM Tris-HCl, 1× proteinase inhibitor) was added and incubated at 4 °C for 30 min with rotation. An 880 μL cell lysate was transferred into a Covaris milliTUBE 1 mL AFA Fiber vial and sheared in a Covaris S220 ultrasonicator (fill level: 10, duty cycle: 5, PIP: 140, cycles/BURST: 200, time: 4 min). Clarified samples were collected by centrifugation at 16,100 rcf for 15 min at 4 °C. After preclearing using 20 μL of protein G beads (Invitrogen, 10003D), 3 μL of anti-AR antibody was added (Abcam, ab108341) for immunoprecipitation overnight. To bind the anti-AR antibody, 60 μL of protein G beads was added and incubated with rotation for two hours at 4 °C. The beads were washed twice each with Low Salt Wash Buffer, High Salt Wash Buffer and LiCl Wash Buffer and resuspended in 100 μL of freshly prepared DNA Elution Buffer (50 mM $NaHCO_3$ and 1% SDS). The ChIP sample beads were placed on a magnet, and the supernatant was collected into a new tube. The above elution step was repeated with another 100 μL volume of elution buffer. The samples were then digested with 10 μL of Proteinase K (Invitrogen, 25530049) with incubation at 67 °C for 4 h. DNA was purified with DNA Clean & Concentrator™-5 (Zymo Research, D4004). One nanogram of eluted DNA was used as input for library construction with a TruePrep DNA Library Prep Kit V2 for Illumina (Vazyme, TD503). Libraries were sequenced with the Illumina NovaSeq sequencing system (PE 2×150 bp reads) at Berry Genomics.

**ChIP-qPCR.** Eluted DNA (0.5 μL) was used as the template for ChIP-qPCR with SYBR qPCR Mix (NovaBio,Q204) following the manufacturer's protocol. The primer sequences used for ChIP-qPCR are listed in Supplementary Data 3.

**ChIP-seq data processing and analysis.** Raw fastq files were first trimmed to remove adaptors using TrimGalore-0.5.0 with the following parameter settings: -q 25 --phred33 --length 35 -e 0.1 --stringency 4. Trimmed fastq files were then mapped to hg19 genome utilizing Bowtie2[41]. Sambamba_v0.6.6 was conducted to remove duplicates[42]. For IGV-2.3 visualization, deepTools-3.2.1 was then performed using function bamCoverage to generate normalized CPM.bw files[43]. For peak calling, MACS2-2.1.1 was utilized with -q 0.05 parameter setting. DeepTools was further applied for heatmap visualization with the function of computeMatrix and plotHeatmap.

**ATAC-seq library preparation.** To reduce the amount of contaminating mitochondrial DNA, we performed an optimized ATAC-seq protocol[44]. In brief, 50,000 cells were collected and washed once with PBS. Cells were then lysed in 50 μL of ice-cold lysis buffer (10 mM Tris-HCl, pH 7.4; 10 mM NaCl; 3 mM $MgCl_2$; 0.1% NP-40; 0.1% Tween 20; and 0.01% digitonin) for 3 min on ice. Immediately after lysis, nuclei were washed with 1 mL of wash buffer (10 mM Tris-HCl, pH 7.4; 10 mM NaCl; 3 mM $MgCl_2$; and 0.1% Tween 20) and then centrifuged at 500 g for 10 min at 4 °C. To prepare sequencing libraries, a TruePrep DNA Library Prep Kit V2 for Illumina (Vazyme, TD501) was utilized for the following steps.

**ATAC-seq data processing and analysis.** The approach used or ATAC-seq data processing was quite similar to that used for ChIP-seq data processing. However, the peak calling step differed due to the lack of input control files. In brief, after raw reads were trimmed with TrimGalore-0.5.0, Bowtie2 was used for mapping the reads to the hg19 genome[41]. Samtools-1.4 was further utilized for bam file sorting and indexing. The bamCoverage function in deepTools was used to generate.bw files with counts per million (CPM) normalization[43]. R-3.6.1 package Diffbind-2.12.0 was used to identify overlapped peaks between AR ChIP-seq-generated peaks in LNCaP cells and ATAC-seq-generated peaks in all four cell lines, respectively. Then, both-open peaks were defined by overlapping the above-generated peaks in all four cell lines. To further identify specific prostate-open peaks, we employed the intersect function in bedtools-2.25.0[45] to exclude peaks that emerged in any of the three lung cell lines in LNCaP cells.

**GSEA analysis**. We downloaded gene expression matrices of multiple cancer cell lines from cBioPortal (http://www.cbioportal.org)[46,47] (derived from cancer cell line encyclopedia). We next performed GSEA to determine whether hallmark androgen response genes show significant differences between AR-positive prostate cancer cells (LNCaP, VCaP, and 22RV1) and AR-positive lung cancer cells (A549, H1437, and H2126)[48]. In addition, we also compared sum of z-scores for hallmark androgen response genes between these two groups.

**Single-cell RNA-seq analysis**. Single-cell RNA-seq analysis was performed on data from the COVID-19 Cell Atlas (https://www.covid19cellatlas.org). To evaluate the mRNA expression levels of target genes in different lung cell types, we selected a single-cell RNA-seq dataset generated from healthy human lungs[25].

**Statistics and reproducibility**. Two-tailed Student's $t$-tests and Mann–Whitney test were performed in GraphPad Prism 7 to compare differences between two groups. One-way ANOVA tests were conducted for comparisons among more than two groups and Tukey's tests were further performed for multiple comparisons in GraphPad Prism 7. All data are presented as the mean ± standard error of the mean (SEM) values. Pearson correlation analysis was performed with the cor.test function in R to identify whether the mRNA expression of one gene was significantly correlated with that of another gene. At least three times each experiment was repeated independently with similar results.

**Reporting summary**. Further information on research design is available in the Nature Research Reporting Summary linked to this article.

## Data availability

The raw data and processed data for ChIP-seq and ATAC-seq data are deposited in the Gene Expression Omnibus (GEO) database under GSE163623 (ChIP-seq) and GSE163624 (ATAC-seq), respectively. These data also have been deposited in NODE (http://www.biosino.org/node) under OEX010556 (ChIP-seq) and OEX010557 (ATAC-seq). Public ChIP-seq and RNA-seq datasets used in this study are available from GEO under the following accession code: GSE148277 (ER ChIP-seq in T47D cells), GSE72249[49] (ER ChIP-seq in MCF7 cells), GSE148276 and GSE148276 (RNA-seq in T47D and MCF7 cells). Source data are provided with this paper.

## Code availability

The code used for processing and analyzing the data in this study is available in GitHub repository: [https://github.com/lifei176/ChIP-seq-ATAC-seq][50].

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

## Acknowledgements
We thank members of the Core Facility of Microbiology and Parasitology (SHMC) and the Biosafety Level 3 Laboratory at Shanghai Medical College of Fudan University, especially Qian Wang, Chengjian Gu. We thank the Genome Tagging Project (GTP) Center, Shanghai Institute of Biochemistry and Cell Biology (CAS), for technical support. We thank Lingling Chen and Yang Wang for providing thoughtful suggestions and technical support in immunofluorescence staining. This study was supported by grants from the Strategic Priority Research Program of the Chinese Academy of Sciences (XDA16020905 and XDB19000000), the National Key Research and Development Program of China (No. 2017YFA0505500), the National Natural Science Foundation of China (81830054, 81772723, and 81822045), the State Key Project for Infectious Diseases (2018ZX10302207-004-002), the Development Programs for COVID-19 of Shanghai Science and Technology Commission (20431900401) and Program of Shanghai Academic/Technology Research Leader (20XD1420300).

## Author contributions
D.G. conceived and designed the experimental approach. F.L., M.H., P.F.D., J.H., and X.T.T. performed most experiments. W.X., Y.W., X.C., Y.T.L., and D.Q. performed the experiments in Biosafety Level 3 Laboratory. S.B.J., Y.H.X., and L.L. supervised and designed experiments in Biosafety Level 3 Laboratory. X.Y.T., X.Y.X, W.X.G., Y.J.Z., Y.G.L., Y.Q.Z, X.Y.Z., Z.L., R.A., S.B.J., Q.W., and H.B.J. helped with the experiments and provided technical support. J.H., M.H., P.F.D., and Q.W. edited the paper. F.L. and D.G. wrote the manuscript. D.G., L.L., Y.H.S., and Y.H.X. prepared the manuscript as the senior authors.

## Competing interests
The authors declare no competing interests.
