## [Peer Review File · Nature Communications]

REVIEWER COMMENTS

Reviewer #1 (Remarks to the Author):

The cellular serine protease TMPRSS2 activates the SARS-CoV-2 spike protein and its activity is essential for SARS-CoV-2 entry into human lung cells in vitro. Moreover, TMPRSS2 is known to be expressed in prostate in an androgen dependent manner and it has been suggested that anti-androgen receptor (AR) agents might block SARS-CoV-2 infection of lung cells by reducing TMPRSS2 expression. However, it has so far not been addressed whether TMPRSS2 is required for SARS-CoV-2 infection of lung tissue and whether AR inhibitors block infection. Li and colleagues show that TMPRSS2 expression is required for efficient SARS-CoV-2 infection of lung tissue in a mouse model. Moreover, they show that the AR inhibitor enzalutamide blocks TMPRSS2 expression and SARS-CoV-2 entry into prostate cells but fails to downregulate TMPRSS2 and to block SARS-CoV-2 entry into lung cells in vitro and in a mouse model. Finally, lack of AR dependent TMPRSS2 expression in lung cells is linked to insufficient accessibility/presence of AR binding sites in the chromatin of lung cells. These results are topical and of high interest to a broad spectrum of researchers. The authors might want to consider some minor points.

Major

None

Minor

Viral load should ideally be shown in figure 1.

Ideally, camostat should have been included in figures 2e and 3k.

The effect of camostat on viral entry into TMPRSS2-positive cell lines was clear but maybe not as high as some readers might expect. Therefore, the authors might want to mention in the text that some of the cell lines examined might express cathepsin L in conjunction with TMPRSS2, which reduces TMPRSS2-dependency of viral entry.

“new worldwide pandemic” a pandemic is by definition worldwide.

“threat due to its high infectivity” should read “threat due to its high transmissibility”

“role of TMPRSS2 in provoking coronaviruses” should read “role of TMPRSS2 in promoting coronaviruses”

Reviewer #2 (Remarks to the Author):

There is great interest in understanding predisposition to COVID-19 infections. One important observation has been that men are more likely to die from COVID-19 and some suggestion that because the androgen regulated gene TMPRSS2 plays a facilitating role in viral infection-- along with ACE2-- that modulating TMPRSS2 with antiandrogen therapy would be beneficial and could reduce infection. Therefore, it make sense to test this hypothesis in the lung. Li et al. have taken on this challenge and have seen no association. While this is a negative study, these negative findings have significant implications. Clinical trials are being planned without strong biologic basis. The use of antiandrogen therapy could have other negative effects related to the immune system

among others. Therefore, these findings are timely and critical to share with the research community.

Comments

1) The authors demonstrate nicely that AR is not regulating TMRPSS2 in the lung. ER has been proposed as a potential mechanism of TMRPSS2 regulation as well as AR. Is there any support that ER modulation would down regulate TMRPSS2 in the lung or facilitate infection?

Minor comments

Line 5: "a relatively new antiandrogen agent". Enza is quite established at this point-would strike "relatively new"

Line 12 and elsewhere-grammatical issues should be addressed. For example "But not neither"

Reviewer #3 (Remarks to the Author):

This is a review of the manuscript entitled, "Distinct mechanisms for TMRPSS2 expression explain organ-specific inhibition of SARS-CoV-2 infection by enzalutamide" by Li, et al. It is generally well written, albeit some figure legend could be made clearer and transparent for the reader.

Major comments:

The manuscript describes studies on the use of TMRPSS2 inhibitor enzalutamide and efficacy against SARS-CoV2 in various tissues/cells - both in vitro and an in vivo mouse model. The authors employ the use of a mouse model to study AR and TMRPSS2. They use a cross of mice (T2Y) to study TMRPSS1 regulated expression by tamoxifen administration - yet there is an absence of tamoxifen neg and tamoxifen pos tissues to compare the validate the YFP IHC in the model. In lieu of Tamoxifen neg mice IHC, it appears HE slides were used and this is not a meaningful substitution. The study of lung infection and lesion development is important, but the validation of the model would be greatly enhanced by corroborating these changes with other approaches. E.g. Fig 1 D - BAL or masked tissue scoring could be used to corroborative inflammation and lesion data to validate group specific changes (PMID: 28812529). While quantification of infected cells is good, IHC detection of cellular infection in a lung model (eg Fig 1 F, G) can be greatly influence by sampling, viral deposition and low N/ group especially when low detection levels are present ~1.2% vs 0.3% of cells. Therefore, corroborating these findings by say showing differences of viral titers in the lungs would facilitate more confidence in the comparisons. The manuscript suggests that AR and TMRPSS2 distribution in the lungs are a rationale as to why enzalutamide would not work to prevent SARS-CoV2 infection in lungs. Immunofluorescence of human lung showing TMRPSS2 and AR cellular protein localization is lacking, yet describing and quantifying the colocalization would be very useful to clearly demonstrate this concept with actual protein data.

Minor comments:

Results 2nd paragraph - "...marked effects on SARS-CoV-2 infection ..." while statistically significantly, I would suggest removing the word "marked" as ~1.2 vs 0.3% are both very low infection rates.

Figure 1. The images in Fig.1 are arrange such that the reader can easily think the top panels of Fig 1B is WT and the bottom is TMRPSS2 ko mice. I would suggest a Y axis label on Fig 1B to clarify these mice.

Fig. 2D, E – Clarify in legend if Enza treatment means "Enza + vehicle"?

Fig 4g. Figure legend says mice with castration and enzalutamide is top image and control is bottom – but this doesn't match the images.

Supplemental fig 1. The figure needs to include Tamoxifen neg samples with YFP IHC on the top panels instead of HE to have value in showing the efficacy of the tamoxifen administration.

Supplemental fig 6. There is minimal reproducible evidence in this figure that there is "blocking of nuclear translocation of AR in prostates of enzalutamide treated castrated mice". This should be deleted or replaced with a figure that validates this interpretation.

Sometimes the error bars in graphs are so big one cannot see all the data points – this is especially noticed when small datasets are graphed (E.g. Fig. 1g)

Point-by-point response to Reviewers' comments and concerns

November 23rd, 2020

RE: NCOMMS-20-37388

Distinct mechanisms for TMPRSS2 expression explain organ-specific inhibition of SARS-CoV-2 infection by enzalutamide

Summary statement. We are grateful to the Editor and Reviewers for their thoughtful comments which pointed out four key issues: 1) identification for the role of ER in modulating TMPRSS2 expression; 2) multiple approaches to further confirm the differences of lung pathology between WT mice and Tmprss2-KO mice challenged by authentic SARS-CoV-2; 3) examination for YFP expression in multiple organs of T2Y mice without tamoxifen administration; 4) verification and quantification for the co-localization of AR and TMPRSS2 in human lungs. Briefly, we have supplied additional data in our manuscript and made the following changes:

- Demonstrated ER negatively regulate TMPRSS2 expression potentially through direct binding in human breast cells, however, such regulation was lacking in human lung cells.
- Corroborated Tmprss2-KO mice exhibited significantly less inflammation, tissue damage and viral loads in lungs when compared to WT mice, which was based on multiple approaches including masked H&E scoring, immune cell infiltration quantification, viral loads quantification and expression survey of inflammatory cytokines and chemokines.
- Validated the lack of YFP expression in the organs of T2Y mice without tamoxifen administration.
- Consolidated and quantified the co-localization of TMPRSS2 and AR in human lungs.
- Applied proper significant tests and corrected grammatical issues and errors in figure legends.

Please see below for the detailed responses to each Reviewer's comments and concerns.

Reviewer #1

The cellular serine protease TMPRSS2 activates the SARS-CoV-2 spike protein and its activity is essential for SARS-CoV-2 entry into human lung cells in vitro. Moreover, TMPRSS2 is known to be expressed in prostate in an androgen dependent manner and it has been suggested that anti-androgen receptor (AR) agents might block SARS-CoV-2 infection of lung cells by reducing TMPRSS2 expression. However, it has so far not been addressed whether TMPRSS2 is required for SARS-CoV-2 infection of lung tissue and whether AR inhibitors block infection. Li and colleagues show that TMPRSS2 expression is required for efficient SARS-CoV-2 infection of lung tissue in a mouse model. Moreover, they show that the AR inhibitor enzalutamide blocks TMPRSS2 expression and SARS-CoV-2 entry into prostate cells but fails to downregulate TMPRSS2 and to block SARS-CoV-2 entry into lung cells in vitro and in a mouse model. Finally, lack of AR dependent TMPRSS2 expression in lung cells is linked to insufficient

accessibility/presence of AR binding sites in the chromatin of lung cells. These results are topical and of high interest to a broad spectrum of researchers. The authors might want to consider some minor points.

We are grateful to Reviewer #1 for providing these thoughtful suggestions to improve our study. We have carefully addressed all the concerns raised here.

Major comments:

None

Minor comments:

1. *Viral load should ideally be shown in figure 1.*

We have supplemented both body weight changes (Fig. 1d) and viral loads (Fig. 1e) into figure 1 in our revised manuscript. In accordance with results from S protein staining, compared to WT mice, Tmprss2-KO mice exhibited significantly lower levels of viral loads in lungs.

2. *Ideally, camostat should have been included in figures 2e and 3k.*

We have revised these figures by including camostat group (Fig. 2d, e and Fig. 3k). We have demonstrated that 10 μ M camostat was capable of inhibiting pseudoviral SARS-CoV-2-S-driven entry (Fig. 2a, b and Fig. 3h, i, j). Intriguingly, 10 μ M camostat only slightly but not significantly inhibited authentic SARS-CoV-2 infection in LNCaP cells and LuOs. Furthermore, we found that 100 μ M camostat significantly inhibited authentic SARS-CoV-2 infection in LNCaP cells and LuOs.

3. *The effect of camostat on viral entry into TMPRSS2-positive cell lines was clear but maybe not as high as some readers might expect. Therefore, the authors might want to mention in the text that some of the cell lines examined might express cathepsin L in conjunction with TMPRSS2, which reduces TMPRSS2-dependency of viral entry.*

We thank Reviewer #1 for this constructive comment. Camostat exhibited less efficacy in preventing SARS-CoV-2-S-driven entry into H2126 cells when compared to Calu3 cells. Since previous studies have demonstrated SARS-CoV-2 is also capable of entering host cells dependent of other than TMPRSS2, we next sought to survey the expression of other factors mediating SARS-CoV-2 infection, including CSTB (cathepsin B), CSTL (cathepsin L) and FURIN (Shang J et al., Proc Natl Acad Sci U S A, 2020, PMID:3237663; Hoffmann M., Cell, 2020, PMID:32142651). Expression analysis from CCLE (<https://portals.broadinstitute.org/ccle/>) revealed remarkably higher expression levels of CTSL and FURIN in H2126 than that in Calu3, which potentially explained cell-specific efficacy of camostat (Supplemental Fig. 5l).

4. *“new worldwide pandemic” a pandemic is by definition worldwide.*

We appreciate Reviewer #1 for pointing out such error in our manuscript. We have corrected this error in our revised manuscript.

5. *“threat due to its high infectivity” should read “threat due to its high transmissibility”.*

We have corrected “infectivity” to “transmissibility”.

6. "role of TMPRSS2 in provoking coronaviruses" should read "role of TMPRSS2 in promoting coronaviruses".

We have corrected "provoking" to "promoting".

Reviewer #2

There is great interest in understanding predisposition to COVID-19 infections. One important observation has been that men are more likely to die from COVID-19 and some suggestion that because the androgen regulated gene TMPRSS2 plays a facilitating role in viral infection-- along with ACE2-- that modulating TMPRSS2 with antiandrogen therapy would be beneficial and could reduce infection. Therefore, it make sense to test this hypothesis in the lung. Li et al. have taken on this challenge and have seen no association. While this is a negative study, these negative findings have significant implications. Clinical trials are being planned without strong biologic basis. The use of antiandrogen therapy could have other negative effects related to the immune system among others. Therefore, these findings are timely and critical to share with the research community.

We thank Reviewer #2 for providing these helpful suggestions and critical concerns. In our revised manuscript, we have demonstrated organ-specific function of ER in modulating TMPRSS2 expression in human breast cells and lung cells, potentially indicating that modulation of ER activity to reduce TMPRSS2 expression would not be a curative therapy for treating COVID19. Additionally, we also carefully corrected improper description in our manuscript.

Major comments:

1) *The authors demonstrate nicely that AR is not regulating TMRPSS2 in the lung. ER has been proposed as a potential mechanism of TMRPSS2 regulation as well as AR. Is there any support that ER modulation would down regulate TMRPSS2 in the lung or facilitate infection?*

We appreciate Reviewer #2 for raising such constructive suggestion which comprehensively strengthen our manuscript. We also agree with Reviewer #2 that ER (estrogen receptor/oestrogen receptor) has been postulated as a potential transcription factor to regulate TMPRSS2 expression (Stopsack KH et al., Cancer Discov, 2020, PMID:32276929). However, whether ER authentically regulates TMPRSS2 expression remains unclear. We have carefully characterized ER function in regulating TMPRSS2 expression.

1. Human lungs are characterized with higher *ESR1* expression than *ESR2*. Since there are two functionally distinct ERs, ER-alpha (encoded by *ESR1* gene) and ER-beta (encoded by *ESR2* gene) (Risbridger et al., Nat Rev Cancer, 2010, PMID:20147902; Levin ER et al., Nat Rev Mol Cell Biol, 2016, PMID:27729652), we first surveyed *ESR1* and *ESR2* expression to distinguish which form is more predominant in human lungs. In accordance with most of other normal tissues and tumor tissues, through analyses on TCGA datasets, we observed that both normal lung tissues and lung tumor tissues expressed higher mRNA expression of *ESR1* than *ESR2* (Supplementary Fig. 8a). We next sought to survey *ESR1* mRNA expression across lung cancer cell lines, and identified H2126 and H1437 with *ESR1* high expression, which had been exploited in our present study (Supplementary Fig.

8b). Furthermore, we also confirmed these two cell lines with ER-alpha protein expression (Supplementary Fig. 8c).

- 2. ER negatively regulates TMPRSS2 expression in human breast cells.** To further verify whether TMPRSS2 expression could be regulated by ER, we treated H2126, H1437 and MCF7, an ER-positive breast cell lines with ER ligand, estradiol (E2) and ER inhibitor, fulvestrant, respectively. Consistent with previous studies (Bahreini Amir et al., Breast Cancer Res, 2017, PMID:28535794; Ghosh M G et al., Cancer Res, 2000, PMID:11103799), mRNA expression of *GREB1*, a well-known ER target gene in breast cells, was drastically increased upon E2 stimulation and concordantly decreased upon fulvestrant treatment in MCF7 cells (Supplementary Fig. 8d,e). Intriguingly, E2 stimulation reduced *TMPRSS2* mRNA expression Supplementary Fig. 8f). On the contrary, fulvestrant treatment promoted *TMPRSS2* mRNA expression in MCF7 cell (Supplementary Fig. 8g).
- 3. ER directly binds to TMPRSS2 in human breast cells.** We also interrogated publicly available datasets of ER ChIP-seq to identify potential mechanisms for ER regulating *TMPRSS2* expression. E2 treatment remarkably promoted ER binding to *GREB1* and enhanced mRNA expression of *GREB1* in both T47D and MCF7 cells (Supplementary Fig. 8h, i, j). In line with our results, E2 stimulation also notably reduced *TMPRSS2* expression (Supplementary Fig. 8l, m). Concordantly, direct ER binding to *TMPRSS2* was observed, which were also sensitive to E2 stimulation (Supplementary Fig. 8k). Hence, these data suggested that ER is capable of modulating *TMPRSS2* expression potentially through direct binding to *TMPRSS2* gene locus in breast cells.
- 4. ER is unable to regulate TMPRSS2 expression in human lung cells.** Distinct from the function of ER in regulating *TMPRSS2* expression in breast cells, neither E2 nor fulvestrant treatment had marked effects on *GREB1* and *TMPRSS2* expression in lung cells H1437 and H2126 (Supplementary Fig. 8d-g). These data indicated that *TMPRSS2* expression was not regulated by ER in ER-positive lung cells, H1437 and H2126.
- 5. The mRNA expression of TMPRSS2 does not significantly correlate to ESR1 in human lungs.** Since we have demonstrated the absence of *TMPRSS2* regulation by ER in two of ER-positive lung cancer cell lines, we next sought to determine whether the expression correlation of *ESR1* and *TMPRSS2* coordinated these findings. Based on freshly dissociated lung cells from 43 normal human lung tissue samples, qRT-PCR analyses on *ESR1* and *TMPRSS2* mRNA expression revealed no significant correlation between these two genes (Supplementary Fig. 8n, o), which was consistent with results from TCGA datasets (Supplementary Fig. 8p).

Collectively, these findings demonstrated ER could negatively regulate *TMPRSS2* expression in breast cells, however, such regulation was lacking in human lung cells, which did not provide supportive evidences for ER activity modulation to reduce *TMPRSS2* expression as a curative therapy to inhibit SARS-CoV-2 infection. In combination with organ-specific activity of AR, further studies to comprehensively identify hormone-responsive organs and the organ-specific function of essential hormones are needed.

Minor comments:

Line 5: “a relatively new antiandrogen agent”. Enza is quite established at this point-would strike “relatively new”

We are grateful to Reviewer #2 for pointing out this inappropriate phrase, and we have corrected this phrase to “a second-generation antiandrogen agent”.

Line 12 and elsewhere-grammatical issues should be addressed. For example “But not neither”

We have carefully checked the manuscript and corrected the errors accordingly.

Reviewer #3

This is a review of the manuscript entitled, "Distinct mechanisms for TMPRSS2 expression explain organ-specific inhibition of SARS-CoV-2 infection by enzalutamide" by Li, et al. It is generally well written, albeit some figure legend could be made clearer and transparent for the reader.

We thank Reviewer #3 for pointing out the following concerns and constructive suggestions. We have corrected the inappropriate description and errors in figure legends to improve our manuscript.

In the revised manuscript, we carefully performed new experiments to corroborate our findings (YFP staining across multiple organs of T2Y mice with or without tamoxifen administration, multiple approaches to evaluate viral loads and lesions in SARS-CoV-2-infected mice lungs, quantification for the co-localization of AR and TMPRSS2 in human lungs, etc).

Major comments:

The manuscript describes studies on the use of TMPRSS2 inhibitor enzalutamide and efficacy against SARS-CoV2 in various tissues/cells - both in vitro and an in vivo mouse model. The authors employ the use of a mouse model to study AR and TMPRSS2. They use a cross of mice (T2Y) to study TMPRSS1 regulated expression by tamoxifen administration – yet there is an absence of tamoxifen neg and tamoxifen pos tissues to compare the validate the YFP IHC in the model. In lieu of Tamoxifen neg mice IHC, it appears HE slides were used and this is not a meaningful substitution.

We thank Reviewer #3 for providing such constructive comments on YFP staining in T2Y mice without tamoxifen administration, which would facilitate demonstrating that YFP expression could specifically indicate Tmprss2-positive cells. To address this question, we carefully examined YFP expression across multiple organs of T2Y mice with or without tamoxifen administration. As expected, predominant YFP expression was only detected in the organs of tamoxifen-administrated T2Y mice (Fig. 1b and Supplementary Fig. 1c). Importantly, YFP-positive organs also comprise YFP-negative cells, which could serve as an internal negative control to demonstrate the cell-specific YFP expression.

Taken together, through surveying YFP expression in multiple organs of T2Y mice with or without tamoxifen administration, we clearly verified specific Tmprss2 expression in multiple organs.

The study of lung infection and lesion development is important, but the validation of the model would be greatly enhanced by corroborating these changes with other approaches. E.g. Fig 1 D – BAL or masked tissue scoring could be used to corroborative inflammation and lesion data to validate group specific changes (PMID: 28812529). While quantification of infected cells is good, IHC detection of cellular infection in a lung model (eg Fig 1 F, G) can be greatly influence by sampling, viral deposition and low N/ group especially when low detection levels are present ~1.2% vs 0.3% of cells. Therefore, corroborating these findings by say showing differences of viral titers in the lungs would facilitate more confidence in the comparisons.

We agree with Reviewer #3 that multiple approaches to evaluate lung lesions are critical. We also feel great thanks for Reviewer #3 for proposing two helpful approaches to address this question, including BAL (bronchoalveolar lavage) and masked tissue scoring. Since the current working hours for our technician is limited to two hours per day in biosafety level 3 (BSL3) facility, BAL is almost impossible right now. To fully address reviewer's concerns, we have employed other useful methods to corroborate our results, including masked tissue scoring, measurement of viral loads, quantification for CD45-positive cells, survey for mRNA expression of inflammation related cytokines or chemokines.

1. Less viral loads were identified in the lungs of Tmprss2-KO mice challenged by SARS-CoV-2.

To directly compare viral loads of Tmprss2-KO mice lungs with that of WT mice lungs, we performed probe qRT-PCR and observed significantly less viral loads in Tmprss2-KO mice lungs (Fig. 1e). This result coordinated with IHC findings that the percentage of SARS-CoV-2 S protein positive lung cells was significantly lower in Tmprss2-KO mice than that in WT mice.

2. Less immune cell infiltration was consolidated in Tmprss2-KO mice. We first employed Zeiss Axio Scan. Z1 to capture fluorescence pictures for CD45 staining in SARS-CoV-2-infected lungs. HALO 2.3 software was then conducted for auto-counting CD45 positive cells and total cells (DAPI). The percentage of CD45-positive cells in total lung cells was significantly lower in Tmprss2-KO mice than that in WT mice, corroborating that Tmprss2-KO mice exhibited less immune cell infiltration in lungs in response to SARS-CoV-2 infection (Fig. 1j).

3. Tmprss2-KO mice exhibited reduced expression of inflammatory cytokines.

Extensive changes in cytokine profiles tightly correlate with COVID-19 disease severity (Mehta P et al., Lancet, 2020, PMID:32192578; Huang C et al., Lancet, 2020, PMID:31986264; Chua et al., Nat Biotechnol, 2020, PMID: 32591762). Hence, we sought to survey the mRNA expression of inflammatory cytokines. Compared with the lungs of SARS-CoV-2-infected WT mice, in Tmprss2-KO mice, we found significantly lower mRNA expression levels of canonical inflammatory cytokines, including *Il6*, *Cxcl10*, *Ifnb* and *Ifng* (Fig. 1k-n). These data also suggested less inflammatory responses to SARS-CoV-2 infection in the lungs of Tmprss2-KO mice compared to WT mice.

4. Significantly milder lung damage was observed in Tmprss2-KO mice. To assess the degree of lung pathogenesis in WT mice and Tmprss2-KO mice after SARS-CoV-2 infection, we performed masked tissue scoring to calculate semiquantitative H&E score. According to previous studies to

assess lung lesions (Gu et al., Science, 2020, PMID:32732280; Leist et al., Cell, 2020, PMID: 33031744), We modified assessment criteria and conducted histopathological analyses as described in “Histopathological analyses” method part of our revised manuscript. Briefly, H&E scores were assessed by evaluation for bronchiole epithelial damage, degeneration of alveolar cells, edema, hemorrhage, parenchymal wall expansion and inflammatory cells infiltration (Referee Fig. 1). With SARS-CoV-2 infection, in comparison with WT mice, Tmprss2-KO mice exhibited significantly lower levels of lesions in lungs (Fig. 1o).

In addition, we also employed these extra approaches to identify the effects of enzalutamide on lung pathogenesis in response to SARS-CoV-2 infection. Compared to WT mice without enzalutamide treatment, enzalutamide-treated castrated mice exhibited comparable levels of lung lesions and similar

Referee figure 1

a

Evaluation index	H&E Scores				Standards for scoring
	0	1	2	3	
Bronchiole epithelial damage	No	<10%	10%~50%	>50%	Damaged bronchiole ducts/Total bronchiole ducts
Degeneration of alveolar cells	No	<10%	10%~50%	>50%	Degenerated alveolar cells/Total alveolar cells
Edema	No	<10%	10%~50%	>50%	Edema region/Total region outside the blood vessels
Hemorrhage	No	<10%	10%~50%	>50%	Hemorrhage region/Total region
Parenchymal wall expansion	No	<10%	10%~50%	>50%	Expansion region/Total parenchymal region
Inflammatory cells infiltration	No	<10%	10%~50%	>50%	Infiltration region/Total region

b

c

d

↑ Perivascular inflammation ◄ Hemorrhage (---) Parenchymal expansion
 ○ Peribronchial inflammation ◄ Epithelial damage * Edema

Referee Figure 1. (a) Table for assessment criteria including evaluation items, H&E scores and standards for scoring. (b). H&E staining in the lungs of WT mice with SARS-CoV-2 infection (2X). (c) Representative pictures to indicate various lesions in the lungs of WT mice with SARS-CoV-2 infection (40X). (d) Histology key for indicating various lesions.

mRNA expression levels of inflammatory cytokines and chemokines (Fig. 4h, j-m), corroborating the lack of therapeutic efficacy of enzalutamide for SARS-CoV-2 inhibition in mouse models.

In summary, through multiple approaches to collectively analyze viral loads, immune cell infiltration, inflammatory cytokines expression and lung pathogenesis, we demonstrated that *Tmprss2* knockout significantly reduced SARS-CoV-2 viral loads and lesions in lungs, implicating TMPRSS2 as an important role in mediating SARS-CoV-2 infection. Additionally, our results also suggested that enzalutamide treatment would not facilitate the inhibition for SARS-CoV-2 infection.

The manuscript suggests that AR and TMPRSS2 distribution in the lungs are a rationale as to why enzalutamide would not work to prevent SARS-CoV2 infection in lungs. Immunofluorescence of human lung showing TMPRSS2 and AR cellular protein localization is lacking, yet describing and quantifying the colocalization would be very useful to clearly demonstrate this concept with actual protein data.

We are very grateful to Reviewer #3 for providing such helpful suggestions. To address this question, on the basis of multiple normal human lung sections from six patients, we performed co-staining immunofluorescence for AR and TMPRSS2 and made quantification analyses. Consistent with results from public single-cell RNA-sequencing data, AR-positive cells and TMPRSS2-positive cell are broadly distributed in both bronchiolar and alveolar regions. In alveolar regions, the percentage of AR-positive cells and TMPRSS2-positive cells were 11.10% and 6.00% respectively, and 1.05% of cells were identified as TMPRSS2/AR double positive cells (Supplementary Fig. 4m, n). Intriguingly, in accordance with results from analyses on single-cell RNA-sequencing data (Supplementary Fig. 4d, g, i, l), higher percentage of both AR-positive cells and TMPRSS2-positive cells was observed in bronchiolar regions. Quantification analyses revealed that 66.08% and 51.21% of cells were characterized as AR-positive cells and TMPRSS2-positive cells respectively, while, the percentage of TMPRSS2/AR double positive cells was 37.18% (Supplementary Fig. 4m, o). These data clearly demonstrated the co-localization of AR and TMPRSS2 in human lungs, as well as indicating the differential percentage of TMPRSS2/AR double positive cells in bronchiolar regions and alveolar regions.

Minor comments:

1. Results 2nd paragraph – "...marked effects on SARS-CoV-2 infection ..." while statistically significantly, I would suggest removing the word "marked" as ~1.2 vs 0.3% are both very low infection rates.

We have removed the word "marked" in our revised manuscript.

2. Figure 1. The images in Fig.1 are arranged such that the reader can easily think the top panels of Fig 1B is WT and the bottom is TMPRSS2 ko mice. I would suggest a Y axis label on Fig 1B to clarify these mice.

We sincerely thank the reviewer for providing this suggestion. We have revised Fig 1b by labelling Y axis to depict that the top panel and bottom panel are YFP IHC staining in *Tmprss2*-KO mice without or with tamoxifen administration respectively.

3. Fig. 2D, E – Clarify in legend if Enza treatment means "Enza + vehicle"?

We thank Reviewer #3 for raising this concern. Since enzalutamide powder was dissolved in DMSO as manufacture's instruction (<https://www.selleck.cn/products/MDV3100.html>), "vehicle" here is DMSO. Therefore, "Enza treatment" indeed means enzalutamide dissolved in DMSO. To exclude such confusions, we have accordingly corrected "vehicle" to "DMSO" in this figure.

4. Fig 4g. Figure legend says mice with castration and enzalutamide is top image and control is bottom – but this doesn't match the images.

We have checked and corrected all the figure legends in our revised manuscript.

5. Supplemental fig 1. The figure needs to include Tamoxifen neg samples with YFP IHC on the top panels instead of HE to have value in showing the efficacy of the tamoxifen administration.

We appreciate Reviewer #3 for raising such constructive suggestion. As we have described in our responses to major concern 1 of Reviewer #3, we have revised these data and replaced H&E with YFP IHC staining in multiple organs of T2Y mice without tamoxifen administration (Fig. 1b and Supplementary Fig. 1c).

6. Supplemental fig 6. There is minimal reproducible evidence in this figure that there is "blocking of nuclear translocation of AR in prostates of enzalutamide treated castrated mice". This should be deleted or replaced with a figure that validates this interpretation.

We have replaced this figure with a more presentative figure. In addition, we have supplemented zoomed figures for clearly interpreting our results.

7. Sometimes the error bars in graphs are so big one cannot see all the data points – this is especially noticed when small datasets are graphed (E.g. Fig. 1g)

We have revised all the figures with error bars to guarantee that their data points could be clearly seen.

REVIEWERS' COMMENTS

Reviewer #1 (Remarks to the Author):

The authors have adequately addressed all points raised by this reviewer.

Reviewer #2 (Remarks to the Author):

The authors have addressed all my concerns in their revision.

Reviewer #3 (Remarks to the Author):

Thank you for the detailed revisions. These changes have significantly improved the manuscript.

Point-by-point response to Reviewers' comments and concerns

December 23rd, 2020

RE: NCOMMS-20-37388A

Distinct mechanisms for TMPRSS2 expression explain organ-specific inhibition of SARS-CoV-2 infection by enzalutamide Summary statement. We are very grateful to the Editor for handling our manuscript. We also appreciate very much for Reviewers for their constructive suggestions and positive comments on our manuscript.

Reviewer #1

The authors have adequately addressed all points raised by this reviewer.

We thank Reviewer #1 for raising positive comments and providing constructive suggestions.

Reviewer #2

The authors have addressed all my concerns in their revision.

We are grateful to Reviewer #2 for the positive comments and help in improving our manuscript.

Reviewer #3

Thank you for the detailed revisions. These changes have significantly improved the manuscript.

We thank Reviewer #3 for the positive comments and raising helpful advices.